# A New Look into Cancer—A Review on the Contribution of Vibrational Spectroscopy on Early Diagnosis and Surgery Guidance

**DOI:** 10.3390/cancers13215336

**Published:** 2021-10-24

**Authors:** Adriana P. Mamede, Inês P. Santos, Ana L. M. Batista de Carvalho, Paulo Figueiredo, Maria C. Silva, Mariana V. Tavares, Maria P. M. Marques, Luís A. E. Batista de Carvalho

**Affiliations:** 1“Unidade de I&D Química-Física Molecular” (QFM-UC), Departament of Chemistry, University of Coimbra, 3004-504 Coimbra, Portugal; apm@uc.pt (A.P.M.); ips@uc.pt (I.P.S.); almbc@uc.pt (A.L.M.B.d.C.); mariana.vide.tavares@ipoporto.min-saude.pt (M.V.T.); labc@ci.uc.pt (L.A.E.B.d.C.); 2Pathology Department, Portuguese Institute of Oncology Francisco Gentil (IPOFG), 3000-075 Coimbra, Portugal; pbsf@ipocoimbra.min-saude.pt; 3Surgery Department, Portuguese Institute of Oncology Francisco Gentil (IPOFG), 3000-075 Coimbra, Portugal; 3483@ipocoimbra.min-saude.pt; 4Gynaecology Department, Portuguese Institute of Oncology Francisco Gentil (IPOFG), 4200-072 Porto, Portugal; 5Department of Life Sciences, University of Coimbra, 3000-456 Coimbra, Portugal

**Keywords:** Raman spectroscopy, FTIR spectroscopy, breast cancer, head and neck cancer, alternative and complementary methodologies, diagnosis, surgical margins assessment

## Abstract

**Simple Summary:**

Cancer is a leading cause of death worldwide, with the detection of the disease in its early stages, as well as a correct assessment of the tumour margins, being paramount for a successful recovery. While breast cancer is one of most common types of cancer, head and neck cancer is one of the types of cancer with a lower prognosis and poor aesthetic results. Vibrational spectroscopy detects molecular vibrations, being sensitive to different sample compositions, even when the difference was slight. The use of spectroscopy in biomedicine has been extensively explored, since it allows a broader assessment of the biochemical fingerprint of several diseases. This literature review covers the most recent advances in breast and head and neck cancer early diagnosis and intraoperative margin assessment, through Raman and Fourier transform infrared spectroscopies. The rising field of spectral histopathology was also approached. The authors aimed at expounding in a more concise and simple way the challenges faced by clinicians and how vibrational spectroscopy has evolved to respond to those needs for the two types of cancer with the highest potential for improvement regarding an early diagnosis, surgical margin assessment and histopathology.

**Abstract:**

In 2020, approximately 10 million people died of cancer, rendering this disease the second leading cause of death worldwide. Detecting cancer in its early stages is paramount for patients’ prognosis and survival. Hence, the scientific and medical communities are engaged in improving both therapeutic strategies and diagnostic methodologies, beyond prevention. Optical vibrational spectroscopy has been shown to be an ideal diagnostic method for early cancer diagnosis and surgical margins assessment, as a complement to histopathological analysis. Being highly sensitive, non-invasive and capable of real-time molecular imaging, Raman and Fourier transform infrared (FTIR) spectroscopies give information on the biochemical profile of the tissue under analysis, detecting the metabolic differences between healthy and cancerous portions of the same sample. This constitutes tremendous progress in the field, since the cancer-prompted morphological alterations often occur after the biochemical imbalances in the oncogenic process. Therefore, the early cancer-associated metabolic changes are unnoticed by the histopathologist. Additionally, Raman and FTIR spectroscopies significantly reduce the subjectivity linked to cancer diagnosis. This review focuses on breast and head and neck cancers, their clinical needs and the progress made to date using vibrational spectroscopy as a diagnostic technique prior to surgical intervention and intraoperative margin assessment.

## 1. Introduction

Cancer is a leading cause of death worldwide, it being estimated that approximately 10 million people died of this disease in 2020 [1]. Cancer incidence has increased over the years, and this trend is expected to continue, since cancer is an age-related disease—with an increasing life expectancy, the number of cancer cases is foreseen to increase over time. 

Cancer diagnosis requires the expertise from several fields of knowledge, from biochemical/chemical to clinical. Histopathological analysis, performed on biopsy-collected samples, is the gold standard methodology for cancer diagnosis, and relies on the contribution of molecular pathology and bioengineering.

A wide range of molecular biology methods have been applied in order to accurately identify features associated with malignancy, as a complement to anatomopathological analysis: reverse transcription polymerase chain reaction (RT-PCR) for the detection of active oncogenes (also known as next-generation sequencing, NGS); immunohistochemistry (IHC) for the detection of the expression of cancer-related proteins/glycoproteins; transcriptomic studies to assess the presence of proliferation markers; epithelial-to-mesenchymal transition and stem cell-like markers, as well as overexpressed or mutated cell cycle regulator proteins [2,3,4,5,6].

Imaging techniques, such as ultrasound, resonance magnetic imaging (MRI), positron emission tomography (PET), computed tomography (CT) and X-rays, among others, were substantial in the cancer diagnostic field, delivering physiological, anatomical, and metabolic information of tumours, sometimes simultaneously. 

Final diagnosis and staging characterise tumours according to the American Joint Committee on Cancer (AJCC) based on the morphological alterations of the tissue, tumour size, grade and location [7]. Although the current practice is functional, the cancer diagnosis process is still time-consuming and subjective, which carries risks for the patients as further discussed.

The combined use of histopathology, IHC, molecular pathology and imaging techniques, rather than their sole use, reduces some of the limitations and errors associated with cancer diagnosis [8,9]. However, it is important to be aware that this is not a static disease. Instead, it is a very complex, dynamic and multifactorial pathology. Tumours evolve over time inside the body and respond to changes in their microenvironment [10,11], leading to varied and often unknown molecular alterations which subsequently prompt morphological changes. Hence, it is impossible to cover all the relevant metabolic and genetic variations undergone during the oncogenic cascade, accounting for the multi-clonal morphological heterogenicity of the tissues. 

In an attempt to improve the currently used tools, the scientific community has looked for highly sensitive alternative technologies that can lead to cancer diagnosis at an early stage in a fast and non-invasive way, providing broader and more accurate information. 

Raman and Fourier transform infrared (FTIR) spectroscopy are optical-based vibrational techniques with high sensitivity, non-invasiveness and real-time molecular imaging capability, without the need for dyes or external probes, providing extremely accurate chemical information. Molecules vibrate at specific energies according to the chemical bonds that constitute them and their chemical environment, meaning that it is possible to identify and quantify each component of a mixture, as well as minor differences between samples, using vibrational spectroscopy, even for inhomogeneous biospecimens and conditions (including in vivo/in situ).

To date, Raman and FTIR have provided the biochemical fingerprint of the genome, metabolome and proteome of biological samples under distinct conditions, delivering biochemical images and allowing researchers to detect molecular pathological changes with high accuracy. Their use has been successful for diagnosing diseases such as Alzheimer’s, dementia [12,13], diabetes [14,15], viral infections [16,17,18,19], and cancer [20,21,22,23], or for identifying bacteria and fungi infecting patients (guiding physicians regarding treatment choices) [24,25,26,27].

The present literature review addresses the use of vibrational spectroscopy techniques to diagnose breast, and head and neck cancers at an early stage, and its potential to achieve the adequate surgical treatment, as a complement to the histopathological analysis. The choice of these two types of cancer was based on the imperative need for an early diagnosis and accurate surgical margin assessment in their treatment.

## 2. Methods

The literature search was carried out using Scopus and PubMed. 

The search of the clinical needs was performed for the period from 2015 to 2021, based on the current practices and guidelines. The keywords used for the search were “breast cancer” OR “head and neck cancer” AND “screening” OR “diagnosis” OR “surgical margins assessment” OR “surgical recurrence” AND “histopathological analysis” AND “inaccuracy” OR “subjectivity”.

The search concerning early diagnosis and surgery guidance using vibrational spectroscopy addressed the period from January 2011 to June 2021. The filtering process included at least three of the following keywords: “breast cancer” OR “head and neck cancer” AND “vibrational spectroscopy” OR “Raman” OR “FTIR” AND “early diagnosis” OR “biopsy” OR “spectral cytology” OR “saliva” OR “urine” OR “surgery guidance” OR “surgical margins assessment” OR “FFPE samples” OR “spectral histopathology”.

The inclusion criteria were the use of human samples and the potentiality for a successful application in the clinics. In vitro and animal models were excluded, except those analysed by SESORS on account of their innovative content and promising inclusion in the clinical flow due to the specific use of handheld probes. The research articles comparing sampling and sample handling methodologies were also excluded. 

The sections that cover the early diagnosis of cancer included the analysis of samples that might be integrated in the clinical screening process, namely: blood, saliva and cytological biospecimens. Among these, the research articles with an exclusively qualitative analysis of the spectral data, i.e., with no statistical analysis, were not considered. New methodologies supporting biopsy-guided research were also included in these sections.

The surgical margins assessment sections included the studies performed in both fresh and frozen human samples, focusing on new methodologies that will affect less negatively the current clinical routine. In turn, surgical guidance was tackled in an innovative way, searching for newly developed technologies that may provide a fast and accurate screening of the excised biospecimens’ surface.

## 3. Results

### 3.1. Which Are the Clinical Challenges?

#### 3.1.1. Histopathology

Histopathology is the standard methodology for cancer diagnosis. After tumour surgical biopsy, the samples are formalin-fixed and paraffin-embedded (FFPE) in order to keep the tridimensional features of the tissue and its viability for observation [28]. The specimens are then sectioned, coated on glass slides and stained with haematoxylin and eosin (H&E) for histopathological analysis, which is performed by well-trained experienced medical pathologists [28]. Additionally, molecular pathology studies are performed with the same biopsy material. Apart from being a fairly slow process, it encompasses a high degree of subjectivity, which constitutes its main drawback. 

The anatomopathologists’ interpretations of the biospecimens assist tumour boards (oncologists, surgeons, radiologists, and other specialist) together with the patients’ options in determining the treatment of choice regarding patients’ prognosis. However, due to tumours’ morphological heterogeneity, misidentification and/or misinterpretation are possible, leading to diagnostic errors.

Aiming at assessing the diagnostic accuracy and reproducibility obtained with histopathology for different types of cancer, Brunyé et al. [29], Elmore et al. [30,31,32], Gilks et al. [33], Sangoi et al. [34] and Thunnissen et al. [35] evaluated the inter-observer and/or intra-observer variability of the diagnosis determined for breast cancer, melanocytic lesions, endometrial carcinoma, micropapillary carcinoma of the urinary tract and pulmonary adenocarcinoma. All these studies found high rates of disagreement when diagnosing the intermediate cancer stages between the healthy and invasive categories. The search/observation process, the individual features of the samples, the different thresholds set by each anatomopathologist and personal experience were some of the factors found to be the source for the lack of accuracy and reproducibility associated with histopathology. When identifying tumour markers, the diagnosis accuracy was increased. In sum, diagnostic errors are more prone to occur when examining the ‘grey’ areas of cancer, the development of new methodologies to successfully accomplish this task being paramount to assist histopathology on increasing the diagnosis accuracy.

#### 3.1.2. Breast Cancer

Breast cancer (BC) is the most common cancer worldwide [36], affecting both females and males [37]. The emergence of BC may be the result several factors: the accumulation of genetic mutations throughout the lifetime, family inheritance, environmental factors (e.g., pollution, exposure to heavy metals), hormone exposure (namely to oestrogen), obesity, alcohol consumption and sedentary lifestyle [2,37,38,39,40,41].

BC mortality has decreased over the years owing to national screening programs leading to an early cancer detection, thus leading to fast therapeutic intervention, but the medical community still faces numerous challenges regarding its diagnosis and surgical margin assessment.

##### Breast Cancer Diagnosis

In countries with developed health systems, BC screening consists of a first clinical examination of the breasts and a bilateral mammogram which can yield a significant percentage of false positives [37]. Thus, ultrasonography is the second preferred imaging technique to assist BC screening, followed by biopsy collection in cases where abnormalities are found [37]. Still, mammography and ultrasonography of highly dense breasts may provide dubious or inconclusive results, urging for more complex imaging diagnostic methods such as MRI or tomosynthesis (a digital 3D mammography). The latter provides images with less tissue overlap, thus revealing hidden tumours, especially in highly dense breasts [37,42], but it is not yet used in routine clinical practice. Apart from the limitations already discussed, these methodologies are expensive, do not provide biochemical information on the tumours, and patients are exposed to hazardous ionizing radiation. In fact, the tumour biochemical signature is a critical factor for diagnosis, since it allows one to predict the best treatment approach as well as the patients’ response to treatment and consequent prognosis.

Endocrine receptors for oestrogen (ER), progesterone (PR) and human epidermal growth factor 2 (HER2) with or without the proliferation protein marker Ki67 are the standard biomarkers assessed for BC through IHC methods. ER- and PR-positive cancers suggest a good response to hormone receptor-targeted therapy. In turn, HER2-positive cancers and ER-, PR- and HER2-negative cancers (triple-negative) usually require stronger chemotherapeutic approaches [5,37,43]. In addition, the assessment of Ki67 (an indicator for proliferation) is highly relevant in ER- and PR- positive cancers, since it determines chemosensitivity [37,43]. 

According to Tang et al., 20% of IHC testing for ER, PR and HER2 biomarkers is inaccurate worldwide [5]. This low reliability may be due to the huge variability of IHC assays, in addition to the standardised protocols, as a consequence of: (i) different protocol procedures regarding fixation; (ii) distinct choices of antibodies; and (iii) thresholds for positivity [3,5]. Hence, alternative BC biomarkers have been developed, such as microRNAs [44,45,46,47] and exosomes [48,49,50], as well as new non-invasive diagnostic methodologies, discussed in Section 3.2 and Section 3.3.

##### Breast Cancer Surgical Treatment

Adequate surgical intervention is considered a curative procedure, sine qua non, in BC treatment. Surgery may be the first step in the treatment or after neoadjuvant therapy (chemotherapy/hormone therapy and radiotherapy) [37]. Although mastectomy is a lifesaving procedure, it is often perceived as a mutilation [51], and it carries a tremendous psychological impact for women, strongly affecting the perception of their body and their self-image [51,52,53]. Therefore, breast conserving surgery (BCS) became preferable in cases where the excision of the malignant lesion with adequate margins is possible, as it has good aesthetic outcomes and, when followed by radiation therapy, leads to similar or even better survival rates than mastectomy [54,55]. The drawback with BCS is the identification of the tumour safe margins intraoperatively, which may require re-resection of the tumour if the margins are positive.

Prior to or during surgery, there is an attempt to predict the tumour location and respective margins through imaging techniques and dye marking. Especially in the presence of non-palpable tumours, image-guided breast surgery is the chosen methodology, resorting to techniques such as wire localization and magnetic seed localization [55,56]. For BC, the margins are considered adequate when they are >2 mm away from the tumour, and for invasive carcinoma (IC), the guideline for clear margins is “no ink on tumour” [57].

In order to ensure that all of the tumour is removed, frozen sections and cytological analysis of the surroundings of the resected tissue can be performed intraoperatively in order to avoid a second surgery in the case of positive margins [56]. Cavity shave margin resection is another procedure performed during surgery which, according to Dupont and co-workers [58], reduces the probability of positive margins from 36% (“no shave” group) to 9.7% (“shave” group) in a randomised group of 296 patients diagnosed with stage 0-III BC. The volume of the tissue removed in the “shave” group was also considerably larger than that removed from the “no shave” group [55,58], which obviously carries additional effort regarding oncoplastic surgery.

Intraoperative assessment of the sentinel lymph nodes in BC is another major challenge faced by clinicians. The current practice includes examination through lymphoscintigraphy associated, or not, with the use of a blue dye and subsequent cytological or histopathological analysis of frozen sections [37,59]. Magnetic sentinel lymph node (Sentimag) has also emerged over the last few years as a good alternative intraoperative imaging technique for intraoperative lymph node detection [60]. One-step nucleic acid amplification (OSNA) is another method for lymph node metastasis assessment, that is progressively being applied intraoperatively, detecting and quantifying the cytokeratin-19 (CK-19) mRNA, providing accurate and consistent results [61]. 

Despite the efforts made to decrease the need for tumour re-resection, the rates of BC-positive margins are still high (ca. 23%) particularly in DCIS and invasive lobular carcinoma (ILC) [55,62,63], the numbers varying among surgeons (15–40%) and institutions [64,65,66]. In addition to the economic impact, the physical and psychological burden of re-resection for the patient is dramatic, along with a higher probability of postoperative complications and poorer aesthetic results. Hence, improving BC surgical margin assessment is a compelling clinical need.

#### 3.1.3. Head and Neck Cancer

Head and neck cancer (HNC) refers to epithelial malignancies that develop in the nasal or oral cavities, larynx, pharynx, and paranasal sinuses. The malignant lesions occurred in the head and neck are mainly squamous cell carcinomas and can be related to human papillomavirus (HPV) infection, alcohol consumption or tobacco use [67,68,69]. It is estimated that more than 930,000 people were diagnosed with HNC worldwide in 2020 [70]. Roughly 50% of the diagnosed cases have died [70]. HNC is not a very common type of cancer but has poor prognosis, making it important to develop new strategies to improve its diagnosis and treatment.

##### Head and Neck Cancer Diagnosis 

Since there is no screening for HNC (unlike for BC), an early diagnosis depends on identifying and detecting symptoms. A fast medical intervention may be limited by socio-economic and health care access factors [71]. Hence, HNC diagnosis relies on physical examination, imaging techniques such as MRI, CT, X-ray, PET, head and neck endoscopy and biopsy collection [69,72]. As discussed previously, these diagnostic technologies (apart from endoscopy) expose the patients to ionizing radiation, are expensive and lack accuracy, along with being unable to provide biochemical information on the tumours.

Narrow band imaging (NBI) endoscopy has emerged over the years as an alternative to conventional endoscopy, which lacks resolution and contrast, often leading to misidentification and undersampling of malignant lesions. NBI increases the contrast between tissue vasculature and the remaining mucosa thus, allowing clinicians to better visualise small lesions [73]. According to a report by Zhou and colleagues, NBI was shown to be a valuable diagnostic tool, since it was able to diagnose HNC with an overall accuracy of 96% based on 25 previous studies and 6187 lesions [74].

Complementary diagnostic tools for HNC, in particular for oropharyngeal cancer, may include the detection of HPV infected cells through in situ hybridization, by IHC targeting the cell cycle regulator protein p16, PCR for detecting viral DNA and, mRNA coding for the viral oncogenes E6 and E7, in selected cases [69,75,76]. Similarly to BC immunohistochemical biomarkers, HNC biomarking has limitations regarding its reproducibility, sensitivity and accuracy. In turn, in contrast to BC, testing for HPV positivity does not have a prognostic objective, as it is known that HPV-positive HNC patients have a long-term survival rate [51], but it remains unclear whether different treatment strategies for HPV-positive and HPV-negative patients are appropriate [67,77].

##### Head and Neck Cancer Surgical Treatment

Adequate HNC surgery is a curative procedure in the early stages of the disease but faces major challenges regarding surgical margin assessment.

For this type of cancer, a clear margin is obtained when the surface of the resected tissue is > 5 mm from the tumour, depending on the location of the primary tumour, although smaller distances were recently recommended [78]. Intraoperatively, this assessment is performed by visual examination of the resected specimen, and palpation and histopathological analysis of frozen sections [79,80], which is not an easy task when bone margins are to be considered [81]. Buchakjian and co-workers evaluated the prognosis and local recurrence of 406 patients with oral cancer undergoing surgery and found that intraoperative margin assessment from frozen sections from the wound-bed was not an ideal margin predictor [82].

According to Williams, the currently used methods during HNC surgery have several limitations regarding tumour localization, namely: after positive margins following resection; tissue shrinkage, affecting the 5 mm margin of clearance; and when sampling the wound-bed, the collected samples may not represent the true tumour area [81].

Head and neck surgery has a tremendous impact on patients’ lives regarding its aesthetic and lifestyle changing outcomes, since the anatomy of oral/nasal cavities and the throat region may be extensively altered. Considerable reconstruction work is usually needed, involving bone and skin autografts [83], which increases the risks for infections and wound-related complications. Hence, an accurate margin assessment is of the utmost importance in order to avoid relapse and reoperation, since the concern regarding loss of function is particularly significant in HNC surgical treatment.

### 3.2. Raman Spectroscopy

The Raman effect is the consequence of a light-scattering phenomenon caused by the interaction between the electrical component of the light wave and electrical charges inside atoms, that are moving (vibrating) from their equilibrium position through stretching, deformation and torsion motions. When this vibration causes a change in the polarizability of the molecule, the corresponding vibrational mode is detected by Raman techniques.

A monochromatic laser is used as an exciting source to increase the probability of photon scattering (since the Raman scattering is a weak effect). The energies of the vibrational states are specific for each type of chemical bond or functional group: as an example, the chemical entity ${\mathrm{CH}}_{2}$ from lipids, proteins and nucleic acids gives rise to unique Raman signals at 1450, 2850–2875 and 2900–2935 cm^−1^ [84] (each representing a specific vibrational mode). These same signals are environment-sensitive and may undergo slight shifts according to different chemical settings around the ${\mathrm{CH}}_{2}$ moieties. The same principle is applicable for the chemical bonds in DNA, RNA, carbohydrates and all other cellular biochemical components. Therefore, it is possible to assess with high sensitivity the biochemical signature of a tissue, any changes being suggestive of different chemical environments. By coupling Raman and infrared spectroscopies to optical microscopy, it is possible to obtain biochemical information associated with the spatial distribution of the different components within the system, thus allowing one to build combined images representative of chemical composition and structure/morphology (Figure 1) [85].

Several modifications of the Raman technique and its spectrometers were implemented with a view to obtain enhanced signals, which is particularly useful when studying highly heterogeneous biological samples:Surface-enhanced Raman scattering (SERS) takes advantage of the interaction between the sample and a rough nanostructured metal surface to which the molecules are adsorbed, which leads to an enhanced Raman intensity. Both nanotags and metallic surfaces can be used (Figure 2a) [86,87].Spatially offset Raman spectroscopy (SORS), developed in 2005 by Matousek and co-workers [88], allows the detection of penetrating photons into the sample by biasing laterally the detection point from the laser incidence point (Figure 2a) [24,89]. Thus, this methodology provides biochemical information from several layers of tissue. Numerous improvements have been developed, with a view to increase its sensitivity, namely: surface-enhanced spatially offset Raman spectroscopy (SESORS) which couples the signal enhancement of SERS to the depth probing ability of SORS (Figure 2a) [90]; and surface-enhanced spatially offset resonance Raman spectroscopy (SESORRS), that combines resonance Raman (RR) [91] and SORS. These surface-enhanced approaches require the use of functionalised nanoparticles (NPs).Coherent Raman scattering (CRS) microscopy takes advantage of two exciting laser beams. Their energy of excitation is chosen according to specific vibrational modes of the sample that the user wishes to highlight (Figure 2b) [92]. In this way, particular biological components may be probed. Taking the example of the ${\mathrm{CH}}_{2}\;$moieties of lipid constituents in tissues, the spatial distribution of ${\mathrm{CH}}_{2}\;$is obtained.

#### 3.2.1. Early Diagnosis of Breast Cancer by Raman Spectroscopy

Attending to its outstanding abilities regarding specificity and non-invasiveness, Raman spectroscopy has been recognised in the last few decades as a very suitable technique for BC screening. The analysis of blood serum by microRaman is an emerging option for non-invasive diagnosis of breast cancer. Cervo et al. took advantage of the SERS technology to diagnose luminal A BC at different stages (localised malignant lesions and locally advanced cancers with lymph node involvement) using blood serum from 20 healthy individuals, 20 luminal A localised BC patients (pT1N0) and 20 locally advanced luminal A BC patients (pTxN+) [93]. Besides demonstrating the capability to distinguish normal from diseased samples with 90% accuracy, 92% sensitivity and 85% specificity, these authors evaluated the ability of SERS to distinguish localised BC from locally advanced BC, having achieved differentiation with 84% accuracy. Additionally, it was observed that spectra from healthy and locally advanced cancer were more similar, suggesting an identical immune responses. Nargis and co-workers reported the study of blood serum from 18 patients diagnosed with BC, and eight healthy females [94]. The healthy samples were distinguished from the tumourigenic ones with 100% specificity and 99% sensitivity. Additionally, stage 2, 3 and 4 cancers were discriminated with >80% specificity and 90% sensitivity. More recently, these researchers compared the accuracy obtained through microRaman and SERS analysis of 29 serum specimens, collected from 17 BC patients and 12 healthy individuals, aiming at discriminating between stage 2, 3 and 4 [95]. Despite the limited number of samples, both methodologies achieved a successful discrimination between cancer stages, with SERS attaining 90% sensitivity and 98.4% specificity, while microRaman yielded 88.2% sensitivity and 97.7% specificity. Moisoiu et al. tested the possibility to differentiate different types of cancers (breast, lung, colorectal, oral and ovarian) through the analysis of blood serum by SERS [96]. A total of 253 individuals were included in this research: 39 healthy volunteers, 42 BC patients, 109 colorectal cancer patients, 33 lung cancer patients, 17 oral cancer patients and 13 ovarian cancer patients. Breast cancer was discriminated from healthy samples with 93.7% sensitivity and 93.6% specificity, with an accuracy of 76% regarding differentiation from other types of cancer.

Analysing urine through SERS, Moisoiu and colleagues gathered 53 female BC patients undergoing mastectomy/lumpectomy and 22 healthy subjects, having attained 95% specificity, 81% sensitivity and 88% overall accuracy [97]. Lin et al. combined affinity chromatography with SERS for the analysis of urine in order to distinguish BC from other cancer types (namely gastric cancer): breast cancer was differentiated from gastric cancer with 82% sensitivity and 90.7% specificity, while breast cancer was distinguished from healthy individuals with 76.5% sensitivity and 87.5% specificity [98].

Beyond the application of microRaman spectroscopy to discriminate between healthy and diseased tissue, two main goals are envisaged: (1) to develop instrumentation that will allow researchers to acquire in depth spectral data—through the skin, mammary glands and adipose tissue—until the suspicious mass is reached by the laser beam, allowing data collection; or (2) to integrate the Raman spectral acquisition with the biopsy procedure, e.g., with the use of a thin needle coupled to a fibre-optic probe, thus allowing researchers to instantly reach a diagnosis, or ensuring that a significant amount of tissue with cancer-related biochemical changes is collected for diagnosis. Regarding the first objective, Nicolson and collaborators applied SESORRS to multicellular tumour spheroids of human breast, using three different nanotags, either isolated or combined, having been able to detect 3D tumours through a 10 mm barrier of porcine tissue [99]. In a similar study, Nicolson et al. successfully detected breast tumour models at 15 mm depth in porcine tissue using one nanotag at a time [100]. In both studies, colour 2D heatmaps were obtained and the precise location of the tumours was attained [99,100]. Regarding in vivo studies, the same authors used an identical methodology to analyse glioblastoma multiforme (GBM) tumours, through the skull, in animal models, using integrin-targeted nanoparticles to induce the SERS effect (Figure 3) [101]. These results were validated through both histopathology and MRI, evidencing the great potential of SESORRS as a non-invasive BC diagnosis technique. 

At present, the main disadvantage of these subsurface Raman approaches is their short penetration depth into the sample (e.g., tissue). The largest depth reported to date was 50 mm in porcine muscle [102], achieved with a transmission geometry using a benchtop Raman spectrometer and not a handheld and back-scattering instrument (in contrast to the studies performed by Nicolson et al. [99,100,101]). There are several possible instrumentation modifications able to enhance the Raman signal and increase the maximum probing depth into the tissue, such as: using higher laser powers; lowering the spectral resolution (in applications where resolution is not essential); or improving the collection efficiency of the spectrograph [24]. In addition, some other issues must be taken into account: (i) the use of nanoparticles (in both SESORS and SESORRS) needs further research, in order to ensure its safe use in humans; (ii) the laser power must be tested in order to avoid tissue damage; and (iii) the instrumentation suitable for in situ application in a clinical setting should be developed, which may be a challenging and expensive task. Nonetheless, the results obtained so far are quite promising and represent a step further for faster and more precise cancer diagnosis.

Regarding the second goal—incorporation of Raman analysis into core needle biopsies—the depth of the analysis can reach 1–2 mm. Desroches et al. developed this technology for brain core needle biopsies, having adapted a commercial core needle to a set of optical fibres that collected the Raman signal (in the high wavenumber region, 2800–3050 cm^−1^) in the tissue surrounding the needle before the sample was collected (Figure 4) [103]. This methodology allowed researchers to perform an in situ intraoperative diagnosis of glioma in 19 patients displaying >60% density of cancer cells, with 84% accuracy, 90% specificity and 80% sensitivity. Such a device is expected to have similar good results in BC, leading to a virtually instantaneous diagnosis when the data are analysed by dedicated classification models. Using a different methodological approach, but aiming at the same purpose, Saha and co-workers used an optical fibre probe connected to a portable Raman spectrometer to monitor 159 samples of breast microcalcifications, from 33 patients, within 30 min after excision through needle biopsy [104]. These authors developed a classification algorithm which allowed them to distinguish calcium oxalate rich microcalcifications (type I, associated with benign lesions) from calcium hydroxyapatite rich ones (type II, associated with proliferative lesions) microcalcifications with 77% sensitivity and 97% specificity. Applying an identical approach, Barman et al. developed a new classification algorithm for simultaneously detecting microcalcifications and diagnosing the underlying breast lesions (BC, benign lesions, fibrocystic changes (FCC) or fibroadenomas), with an overall accuracy of 82% [105]. This approach enabled the diagnosis of DCIS, which had never been achieved with previous classification algorithms, thus meeting a long-standing issue in breast cancer management—a reliable diagnosis of DCIS.

#### 3.2.2. Surgical Margins Assessment in Breast Cancer by Raman Spectroscopy

An accurate assessment of tumour margins intraoperatively is still an unmet clinical need. Several studies have been performed aiming at the development of a reliable methodology to achieve this goal, mainly through ex vivo analysis of the tissues after surgical excision. The most important feature of Raman spectroscopy for intraoperative use is its ability for providing label-free and fast results, since long periods of time increase anaesthesia exposure and the risk of infection for the patients. At present, histological evaluation of tumour margins during surgery can take up to 60 min and carries a high degree of subjectivity and inaccuracy (discussed in Section 3.1.1, regarding histopathology challenges).

Taking advantage of the current practice of intraoperative analysis of frozen sections, Koya and collaborators studied 88 samples of healthy breast tissue, luminal and basal BC through microRaman spectroscopy, having achieved (upon application of a deep learning algorithm) a reliable differentiation between normal and cancerous tissue, with 90% accuracy, 88.8% sensitivity, and 90.8% specificity [106]. Based on the signals with the highest contribution discriminating normal from cancerous samples, biochemical maps (false-coloured RGB images) were generated and were found to agree with the hematoxylin and eosin (H&E) stained slides prepared for histopathological evaluation. Kong et al., in turn, reported the study of frozen breast tissue sections collected during BCS in 60 patients, being able to diagnose ductal carcinoma (DC) with 95.6% sensitivity and 96.2% specificity [107].

One of the emerging approaches for BC intraoperative margins assessment is multimodal spectral histopathology (MSH) which is the combined use of autofluorescence (AF) and Raman confocal microscopies. The concept behind MSH is to enable the selection of the areas to be measured with the Raman according to the absence of fluorescence (i.e., the areas of tissue with less fluorescence are the ones to be analysed by Raman spectroscopy). Using this approach, Kong and co-workers successfully diagnosed BC in large areas of resected tissue (5 × 5 mm^2^) in ca. 17 min, a feasible time for the application of this technique in the operation theatre [107]. Similarly, Shipp et al. were able to identify positive margins of whole breast specimens with 4 × 6.5 cm^2^ surfaces, immediately after excision, in 12–24 min, with 95% sensitivity and 82% specificity [108]. Furthermore, invasive carcinoma was discriminated from DCIS on 1 mm^2^ tissue surfaces. Autofluorescence imaging in total internal reflection (TIR-AF) mode was combined with Raman microspectroscopy by Lizio and co-workers, in order to assess the surgical margins of fresh local excisions of large dimensions (up to 10 × 10 cm^2^) obtained from BCS and previously diagnosed as invasive carcinomas (Figure 5) [109]. The authors were able to locate the tumours in the resected specimens in 45 min. This time was reduced by half through the use of a more automated setup (approximately 30 min of the experiments were spent on manually switching the sample between equipment and uploading the images to an in-house-developed algorithm). Although this study was performed solely for invasive carcinomas, the promising results thus obtained may render TIR-AF coupled to microRaman a suitable method for assessing tumour margins within a surgical timespan.

A similar measurement selection methodology was developed by Liao et al., who used a modified Raman microspectrometer to rapidly screen the adipose tissue in the margins of breast tissue sections from mastectomies (ranging from 4 × 6 mm^2^ to 20 × 20 mm^2^), which hardly contain cancerous alterations, therefore largely reducing the spectral acquisition time [110]. The areas identified to be adipose tissue through Raman were in agreement with the analysis of the H&E slides, which validates microRaman as a feasible and suitable methodology for the selection of the sampling points when evaluating the specimen margins. Additionally, it is estimated that the analysis of large specimens (5 × 5 cm^2^) will take 20 to 25 min to be completed, well within the surgical timespan.

In a very different approach, Thomas and collaborators developed a 3D scanner with a Raman probe (Marginbot), in which the samples horizontally rotate while the probe rotates alongside, collecting spectral data from the surfaces of the specimens, reaching 2 mm in depth (Figure 6) [111]. Depth-averaged Raman spectra (based on the SORS concept but with less depth resolution) were collected from five patients undergoing prophylactic mastectomies. Tridimensional images were obtained from a ca. 56 cm^2^ specimen in 7–15 min, achieving a discrimination of fibroadenomatoid from adipose tissue with 93% sensitivity and 85% specificity.

A SORS probe was designed by Keller et al. to assess the surgical margins of partial mastectomies in 35 frozen samples comprising invasive ductal carcinoma (IDC), invasive lobular carcinoma (ILC) and benign tissue [112]. The measurements were performed on sites with healthy as well as tumour-apparent features (later confirmed by pathology), with 95% sensitivity and 100% specificity. Positive margins were detected at 2 mm from the surface in 100% of the malignant cases.

Although less common, the SERS effect was also tested for surgical margin assessment by Wang and co-workers, who targeted the epidermal growth factor (EGF) receptors, HER2, ER, as well as the cell surface glycoprotein CD44 with antibody-functionalised nanoparticles [113]. The NPs were topically applied on fresh specimens collected from 57 patients undergoing either BCS or mastectomy. Raman acquisition took 10–15 min for each sample (scanning 3 cm^2^ per min), yielding images evidencing the distribution of EGFR, HER2, ER and CD44 at the tissue surface, with 89.3% sensitivity and 92.1% specificity. This approach was also able to identify triple-negative BC. However, there are still concerns regarding the use of antibody-targeted nanoparticles, regarding fixation and washing protocols that can lead to false positives. 

One of the biggest challenges faced by surgeons is, as mentioned above, the detection of positive lymph nodes. The potential of Raman microspectroscopy to assess metastasis in sentinel lymph nodes was evaluated by Horsnell and collaborators in surgically removed breast tissue samples, which were frozen-cut prior to analysis [114]. Two data acquisition methodologies were applied, differing in the number of measured points—either 5 or 10. For the latter (10 points), healthy lymph nodes were differentiated from cancerous lymph nodes with 81% sensitivity and 97% specificity, upon application of a support vector machine (SVM) algorithm. More recently, Petterson et al. developed an innovative multifibre Raman probe integrated in a hypodermic needle, which was tested in one human lymph node, providing very good signal-to-noise spectra in just a few seconds [115]. With the proper classification algorithm, this is a promising equipment for intraoperative use aiming at the assessment of sentinel lymph nodes (not only in BC but in other types of cancers as well, namely HNC).

Zúñiga and co-workers approached surgical margin assessment from a very different point of view, proving that widespread Raman spectroscopy for clinical purposes may be faster implemented [116]. Two affordable and portable commercially available Raman devices (for non-medical applications) were tested. The authors claim that over 90% accuracy may be achieved with both devices when differentiating healthy from malignant surgically removed frozen breast sections [116]. Although a limited amount of data were reported, this study shows that the combined knowledge from clinicians, researchers, engineers and the industry will allow the development of good quality and affordable equipment in the near future.

#### 3.2.3. Head and Neck Cancer Early Diagnosis by Raman Spectroscopy

Since there is no screening for HNC, as well as the death rate associated with this type of cancer being high, added to the limitations faced by HNC survivors undergoing extensive reconstruction surgeries, the development of non-invasive and fast diagnostic methodologies is paramount.

Regarding non-invasive diagnosis, Connolly and co-workers reported the study of both saliva and oral cells, collected concomitantly, from 18 healthy people and from 18 oropharyngeal cancer patients, using SERS, attaining 89% sensitivity and 57% specificity when using saliva, and 68% sensitivity and 52% specificity when using oral cells to diagnose oropharyngeal cancer [117]. Falamas and collaborators used the conventional microRaman approach and lyophilised saliva, collected from 19 HNC patients and 13 healthy individuals, in order to determine the spectral biomarkers able to distinguish the healthy from the cancerous samples, finding seven Raman signals with discrimination potential, attaining 83% accuracy [118]. Aiming at the same goal, Sahu et al. collected sera samples from 40 patients with squamous cell carcinoma (SCC) of the tongue and 14 of the buccal mucosa, and from 16 healthy participants, achieving 78% efficiency in discriminating normal from tumour samples [119]. The sensitivity and specificity values obtained in some of these studies were lower than desired for a clinical application, suggesting that serum and saliva do not have enough discriminatory properties for a diagnosis of HNC. Xue and colleagues, in turn, used sera samples from 135 OSCC patients to assess the capability of SERS to classify different stages (T1 to T4) and lymph node involvement (N0 to N2), correctly classifying T1 and T3 with 80% accuracy, T2 with 71.7% accuracy and T4 with 77.8% accuracy. Regarding lymph node involvement accuracies of 75.5%, 93.8% and 92% were achieved for N0, N1 and N2, respectively [120]. 

Brindha et al. used urine to determine whether an early diagnosis of oral cancer using Raman spectrometry was possible. Collecting urine from 93 oral cancer patients and 74 healthy subjects, the authors were able to discriminate the healthy samples from the cancerous ones with 98.6% sensitivity, 87.1% specificity and 93.7% overall accuracy [121]. In another study, the same group used the high wavenumber region of the Raman spectra to probe urine from 80 healthy volunteers, 57 patients with pre-malignant lesions and 60 patients with oral cancer, having achieved 98.7% specificity and 91.9% sensitivity when discriminating healthy from malignant and pre-malignant samples [122]. Comparing the accuracy attained with the fingerprint range and that reached with the high wavenumber region, the latter showed a higher discriminant capability. Jaychandra and co-workers concomitantly tested urine, blood plasma, and saliva from 94 patients diagnosed with leucoplakia and oral submucous fibrosis, 63 oral SCC patients and 48 healthy individuals, reaching a correct classification of normal, malignant and premalignant samples with 90.5% accuracy using urine, 93.1% accuracy using saliva and 78% using blood plasma [123]. Eighty-nine samples of oral biopsy-collected tissues were also monitored (normal, premalignant and malignant), yielding 97.4% accuracy for discrimination based on these three groups of samples.

In a totally different approach, Singh and co-workers coupled a fibre-optic probe to a commercial Raman spectrometer to analyse in vivo 104 subjects with oral cancer, pre-malignant lesions on cancer patients, healthy individuals with and without smoking habits, correctly predicting tumour cases with 86% efficiency, pre-malignant lesions with 72% efficiency and contralateral normal samples with an efficiency of 74% [124]. The test data of healthy controls, healthy tobacco users and pre-malignant samples were correctly predicted with 94%, 86% and 55% efficiency, respectively, higher rates of misclassifications being found between pre-malignant lesions and healthy tobacco users. Later, Krishna and colleagues, using an in-built portable Raman device, performed in vivo measurements on 113 OSCC patients, 25 oral submucous fibrosis patients, 33 patients with leukoplakia (a high-risk dysplasia) and 23 healthy subjects. Healthy individuals were distinguished from the cancer and high-risk patients with 94% sensitivity and specificity, pre-malignant lesions being correctly classified in 88% of the measured sites and malignant lesions in 84% of the sites [125]. More recently, Guze et al. used an existing Raman probe and performed in vivo measurements in 18 patients, mimicking a clinical setting [126]. Oral SCC, inflammation, fungal, mild to severe dysplasia and hyperkeratosis were under study, with data collection taking 5 min and achieving 100% sensitivity and 77% specificity when discriminating benign lesions and healthy tissue from SCC and mild or severe dysplasia. Using a similar approach, Bergholt and colleagues used a previously developed Raman probe—designed for upper gastrointestinal endoscopy [127]—that could be integrated into transnasal endoscopes, creating an image-guided Raman endoscopy [128]. In this study, the spectra from different areas of the head and neck of 23 healthy patients were obtained and characterised.

Aiming at an analysis of samples obtained through biopsy, Vohra and co-workers took advantage of the SERS effect using DNA-functionalised nanorattles, targeting the cytokeratin biomarker RNA, CK14, specific for head and neck squamous cell carcinomas (HNSCC) and micrometastases in lymph nodes [129]. Intraoperatively collected samples from 25 HNSCC, thyroid papillary carcinoma, tonsillar disease, benign lymphoid or lymphoma patients were used for RNA extraction, providing results with 100% sensitivity and 89% specificity. It is important to note that the use of SERS for clinical applications has some limitations. In fact, the use of functionalised antibodies, RNA or DNA carries some degree of inaccuracy and irreproducibility regarding fixation and washing protocols.

Cytological Raman analysis has proved to be a promising option for oral cancer screening. However, the use of cell pellets involves a higher sample heterogeneity due to the collection of several types of cells without a spatial reference, thus requiring spectrum- and patient-wise data analysis. Sahu and co-workers have been working on these approaches for the last few years [130,131,132,133]. In an exploratory work, exfoliated cells from healthy controls (10 smokers and 20 non-smokers), 12 diagnosed with leukoplakia and 9 with tobacco pouch keratosis, were collected from the lesions and contralateral sites [130]. Higher rates of misclassifications were found for the tobacco, leukoplakia, contralateral and tobacco pouch keratosis, due to tobacco use and high heterogeneity of the dysplastic lesions. Later, using the same methodological approach, healthy vs. oral cancer patients were assessed, this time performing patient- and spectrum-wise data analysis, with patient-wise analysis providing a higher true classification [131]. Then, oral premalignant lesions were tested against healthy individuals’ oral cells exfoliations, allowing the authors to conclude that with patient-wise analysis, a sensitivity of 70% was achievable when classifying healthy non-smoker individuals, healthy smokers and patients with premalignant lesions, whereas spectrum-wise analysis provided a sensitivity of 77% [132]. More recently, using the same approach, these authors confirmed previous results hypothesising the misclassifications between tumour and contralateral samples as being attributable to malignancy recurrence in treated patients [133].

Hole and collaborators also collected oral exfoliated cells from 29 oral cancer patients and 15 healthy individuals, with no smoking habits, and found that the patient-wise data analysis approach yielded a classification efficiency of 86% [134]. In a similar study, Ghosh et al. collected cells from the oral mucosa from healthy individuals, with and without smoking habits (*n* = 11), and from patients diagnosed with leukoplakia (*n* = 13) and OSCC (*n* = 10), having obtained an accuracy of 82% and 80% from spectra- and patient-wise analysis, respectively [135].

Behl et al. were able to distinguish healthy donors from oral cancer patients with 94% sensitivity through the analysis of the cells’ nuclei and 86% through the analysis of the cytoplasm [136]. Additionally, the authors assessed the confounding factors associated with misclassifications obtained with Raman analysis of exfoliative cytology, concluding that, unlike in previous works, smoking, alcohol consumption, age and gender do not appear to contribute for classification errors, while the site of sampling was found to be determinant.

#### 3.2.4. Head and Neck Cancer Surgical Margins Assessment by Raman Spectroscopy

In Section 3.1.3, the urgent need for an adequate assessment of the surgical margins in HNC treatment was discussed. Barroso and colleagues have developed extensive work under this objective [137,138,139]. Using a confocal Raman microscope built by the team to acquire spectral information in the high wavenumber region (>2500 cm^−1^), Barroso et al. reported the study of normal and tongue SCC collected through surgery, from 14 patients, finding that tumour areas had a higher water content than the surrounding tissue, being able to distinguish SCC from benign regions with 99% sensitivity and a specificity of 92% [137]. Analyses were performed on fresh specimens, within 30 min after excision, without compromising the sample or the routine of the pathology analysis. In a follow-up study, the same authors were able to identify the tumour borders of 25 specimens, based on the increase in the standard deviation of the water concentration in the vicinity of clear margins (approximately 4 to 6 mm from the tumour)—in the tumour, water concentration was found to be 76% ± 8%, and 56% ± 24% in the surrounding healthy tissue [138]. Later, Barroso et al. assessed the fresh bone resection margins, from the mandible of 22 patients diagnosed with oral SCC, reaching a depth resolution of 40 µm, in <30 min, reaching 95% accuracy, 87% specificity and 95% sensitivity in discriminating bone with a tumour from bone without a tumour based on water concentration calculation (Figure 7) [139]. 

Using an image-guided diagnostic approach, Cals and co-workers reported the study of 25 samples, 11 of which were diagnosed as oral squamous cell carcinoma (OSCC), collected from 10 patients undergoing surgery, using an inverted Raman microscope designed to analyse bacterial samples [140]. Fresh-frozen sections were obtained within 60 min after resection, yielding false-coloured heat maps identifying OSCC over healthy adipose tissue, muscle, and nerves with 97% accuracy. Although connective tissue, glands and the squamous epithelium were most frequently misidentified as tumour structures, connective tissue was correctly identified over OSCC in 93% of the cases, squamous epithelium in 75% and glands in 94%. Using the same equipment, Cals et al. later reported the study of 44 samples of SCC of the tongue, collected through surgery from 21 patients [141]. As in their previous work, colour heatmaps were generated using two different data analysis approaches to discriminate healthy from tumour structures: those obtained with the two-step PCA-hierarchical LDA provided 91% accuracy (100% sensitivity, 78% specificity) over the PCA-LDA with 86% accuracy (100% sensitivity, 66% specificity). Although the acquisition times were too long to be compatible with the operating room, the coloured heatmaps were in agreement with the H&E sections obtained for the pathology analysis.

Likewise, Hoesli and co-workers used CRS microscopy (Section 3.2, Figure 1b) to observe 42 HNC samples and 42 healthy sections adjacent to the tumour, two-coloured images being generated, highlighting the stretching mode of ${\mathrm{CH}}_{2}$ from lipids in the cytoplasm and the ${\mathrm{CH}}_{3}$ stretching mode from proteins and lipids [142]. Sample preparation and acquisition was claimed to have taken 30 s, and the analysis of the CRS images was performed by a trained pathologist: the correspondence between the CRS and H&E images being 88% sensitivity and 95% specificity (Figure 8). This study was presented as a faster alternative to H&E staining of frozen sections for intraoperative margin assessment, since CRS imaging allows the observation of atypical nuclei and hypercellularity. However, this methodology would still depend on the pathologist’s opinion and expertise.

### 3.3. FTIR Spectroscopy

Infrared spectroscopy, as with Raman, provides information on the molecular vibrations, but these two techniques are complementary due to the distinct physical processes underlying each, namely: an infrared signal appears when there is a change in the molecule’s dipole moment during the vibration, while a Raman signal is dependent on the variation of polarizability that occurs during this same vibration. The physical phenomenon of FTIR is light absorption rather than scattering, to which Raman spectroscopy is sensitive. As a consequence, some vibrational modes are seen with FTIR and not with Raman and vice versa. Biological tissues are 70% water, and O-H bonds provide strong signals in infrared spectra, overriding the bands of other biochemical constituents. Therefore, the use of FTIR in fresh tissue samples is not as wide as Raman spectroscopy, because it usually demands more complex equipment or additional sampling and data processing steps. Nonetheless, different equipment geometries have been developed in order to overcome some of the challenges faced by using FTIR in biological samples, such as (Figure 9):The transmission mode, the infrared light travels through the sample, being detected after the interaction with the sample.In the reflection mode, in contrast to transmission, the detected infrared light is that reflected by the sample.In attenuated total reflectance (ATR) mode, a crystal with a high refractive index is placed in close contact with the sample. The infrared light passing through the crystal is totally reflected by the crystal walls after interacting with the sample through an evanescent wave.

#### 3.3.1. Early Diagnosis of Breast Cancer by FTIR Spectroscopy 

The approaches for BC early diagnosis based on FTIR, similarly to those under development using Raman spectroscopy, include both non-invasive methods probing saliva or blood serum and the integration of FTIR into existing diagnostic methodologies. 

Regarding the former, Ferreira and colleagues reported the analysis of saliva samples from 30 patients by means of FTIR-ATR: 10 were diagnosed with breast malignancy, 10 with breast benign lesions and 10 with no breast findings. Despite the limited number of samples, discrimination between BC and benign lesions was achieved with 90% sensitivity and 70% specificity. When distinguishing BC from healthy patients, 90% sensitivity and 80% specificity was reached [143]. Since saliva has a high water content, sample preparation included freezing and lyophilization in order to remove as much water as possible prior to analysis. 

Zelig and coworkers collected blood plasma and peripheral mononuclear cells from 24 BC patients and 26 healthy individuals (15 of them with benign breast lesions), and microFTIR measurements were performed on ZnSe slides, yielding 87% sensitivity and 78% specificity [144]. Blood serum was also tested by Elmi et al., who left 43 BC and 43 healthy samples to dry, at room temperature, on zinc selenide crystal disks (IR-transparent) and then measured through FTIR in transmission mode. Using this method, 92% sensitivity, 85% specificity and 90% accuracy were achieved when discriminating BC from healthy specimens [145]. Sitnikova and co-workers also used blood serum for the diagnosis of 66 BC patients and 80 healthy donors through FTIR-ATR, letting the sera samples to dry directly on the ATR crystal. Using this technique, 92.3% sensitivity and 87.1% specificity were obtained when differentiating healthy from cancerous samples [146].

Concerning the integration of FTIR into conventional diagnostic methods, MRI-guided near-infrared spectroscopy (NIRS, which is a combination of MRI with optical imaging), was reported by Mastanduno and colleagues. This approach allowed the authors to generate MRI-NIRS breast scans, from 44 patients, by quantifying: oxy vs. deoxy-hemoglobin, the scattering parameters of different tissues (tissue optical index (TOI) for, e.g., adipose vs. fribroglandular), and water vs. lipids content [147]. Although some limitations arose from an heterogenous optical coverage of the breasts and the fact that the participants were still exposed to high doses of ionizing radiation, the use of NIRS allowed us to discriminate malignant from benign lesions. 

#### 3.3.2. Surgical Margins Assessment of Breast Cancer by FTIR Spectroscopy 

The use of FTIR spectroscopy in fresh tissues carries additional challenges regarding the water content. In order to overcome this drawback, Zhao and co-workers developed a handheld ATR-hollow optical fibre (ATR-HOF) and connected it to a FTIR spectrometer, which was then applied to the study of articular cartilage, postoperatively (Figure 10) [148]. This device was later used by Lu et al. to study fresh breast samples, collected from 12 patients undergoing BCS, reaching >90% accuracy when discriminating healthy from cancerous areas [149]. The use of an ATR configuration decreased the contribution of water in the lower wavenumber region, with the water effect becoming neglectable. Likewise, Tian and colleagues used a similar ATR hollow fibre probe connected to a FTIR spectrometer in the operating room, to analyse 149 sentinel lymph nodes, collected from 49 BC patients, spending 2 to 3 min per measurement [150]. Normal nodes were successfully distinguished from malignant nodes with 94.7% sensitivity and 90.1% specificity. Moreover, 38 of the analysed lymph nodes contained metastasis, which was confirmed by histopathological analysis. Using the same equipment, Tian et al. probed 100 fresh breast resections collected from 100 patients, particularly in the centre of the lesions (not the margins), having achieved 90% sensitivity and 98% specificity when discriminating normal from cancerous sections [151]. 

In sum, with proper instrumentation and suitable methodologies for intraoperative use, the analysis of fresh tissues through FTIR is feasible and suitable for clinical applications, as it provides accurate results with great discrimination potential.

#### 3.3.3. Early Diagnosis of Head and Neck Cancer by FTIR Spectroscopy

The potential of saliva was also tested for the early diagnosis of HNC with FTIR spectroscopy by Zlotogorski-Hurvitz and co-workers who isolated exosomes from the saliva from 13 healthy individuals and 21 patients with oral cancer, reaching 100% sensitivity, 89% specificity and 95% accuracy discrimination [152]. As performed in previous studies, the samples were left to dry onto the ATR equipment for further analysis. Similarly, Falamas et al. used transmission FTIR, on KBr tablets, to study the saliva of 19 HNC patients and 13 healthy individuals and determine the spectral biomarkers capable of distinguishing healthy from cancerous samples [118]. Three main signals were identified, with the classification for oral SCC achieving an accuracy of 82%.

Rai and colleagues, in turn, analysed blood sera from 30 healthy individuals and 30 patients with oral submucous fibrosis (OSF, a benign lesion with the highest malignant potentiality among all dysplastic lesions). The predictive capability estimated for the PLS-DA model was higher than 90%, providing good discrimination between healthy and OSF samples [153]. 

As previously mentioned in Section 3.2.3, in addition to head and neck early diagnosis through Raman spectroscopy, exfoliative cytology analysis using FTIR has also been performed, although less frequently. Townsend and co-workers collected oesophageal cells, using a standard cytological brush attached to an endoscope, from 9 normal squamous tissue, 12 Barrett’s oesophagus and 5 dysplastic lesions [154]. The analyses were performed using microFTIR, reaching >90% sensitivity, specificity and accuracy when discriminating normal squamous cells from Barrett’s oesophagus and dysplasia. Concomitantly to their Raman cytological analysis, Ghosh et al. tested the discriminatory potential of FTIR exfoliative cytology of oral cancer, and concluded that this methodology was able to differentiate healthy from premalignant and malignant lesions with 84.8% accuracy through a spectrum-wise approach, and 82.3% accuracy via a patient-wise approach [135]. A combined application of Raman and FTIR provided an overall accuracy of 97.7%. Although this study was performed with few samples, it shows that the complementary use of Raman and FTIR delivers a higher discriminatory capacity.

#### 3.3.4. Surgical Margins Assessment of Head and Neck Cancer by FTIR Spectroscopy

As previously discussed, FTIR analysis of fresh specimens is extremely difficult due to the strong contribution of the O-H vibrational modes of the water molecule, which hinders the FTIR spectra of samples with higher contents of water. According to Barroso et al., healthy areas of the oral cavity have 56% water, whereas tumours have 76% [138], and therefore the use of FTIR in HNC intraoperative margin assessment is highly restricted. However, it has provided good discriminatory potential in saliva exosomes and blood serum, being a good option for early non-invasive diagnosis of HNC.

Regarding the early diagnosis of BC and HNC, a lot of work has been done in order to obtain faster and more accurate results along with the development of non-invasive diagnostic methodologies. So far, different approaches using FTIR and Raman spectroscopies were successfully developed and applied, providing highly accurate results. 

Surgical margins assessment requires special equipment so that the spectral analysis can be performed in loco—in the operation room—and rapidly, in a way that is compatible with the operation timespan without additional danger for the patient. To date, the technology has been successfully developed and there are several computed classification algorithms, providing highly accurate, sensitive, and specific results. 

Upon improvement of the spectrometers and the automation of the entire process, both surgical margins assessment and diagnosis using vibrational spectroscopy would be possible in the near future.

### 3.4. Spectral Histopathology

Spectral histopathology is of utmost importance to assist anatomopathologists and increase the precision of the diagnoses obtained through histopathological analysis. Confocal spectral microscopes are used in order to obtain false-coloured maps depicting the distribution of the different biochemical components of the samples. FFPE unstained histological sections are used; although this approach is less common for Raman spectroscopy, it is the preferable methodology for FTIR analysis, since the sectioned tissues are dehydrated, and thus there is no water contamination of the FTIR spectra. The major challenges of FTIR histopathology are the substrates, that must be compatible with the FTIR analysis. For a complete spectral acquisition, in the transmission mode, the samples must be mounted on CaF_2_ or BaF_2_ slides, which are very fragile and expensive for routine use. In fact, the use of standard histological slides hinders the acquisition of the spectra below 2200 cm^−1^, since glass absorbs infrared light. Furthermore, in reflection mode, the quality of the spectra is highly dependent on a perfect focus on the sample, which may be problematic due to the irregularity of the tissue sections, and a point-by-point focus would considerably increase acquisition times.

Finally, the use of FFPE samples carries additional work regarding sample manipulation and data processing, since paraffin has a strong spectral contribution, overriding the tissue signals. Hence, chemical [155,156,157,158] and digital dewaxing [159,160,161] are frequently needed prior to data processing and analysis.

#### 3.4.1. Spectral Histopathology of the Breast

Raman histopathology, performed in confocal Raman microscopes, was used by Vanna and co-workers, who reported the study of breast microcalcifications from 56 patients undergoing core biopsy, having attained 93.5% sensitivity and 80.6% specificity when discriminating malignant from healthy samples (including those out of the lesion area) [162]. Similarly, Lyng and collaborators analysed breast tissue samples, from 20 patients, representative of benign (fibroadenoma and fibrocystic lesions, intraductal papilloma) and cancerous (IDC and lobular carcinoma) lesions, being able to differentiate the benign from the malignant areas using a combination of different chemometric methods, with the best classifier having achieved >90% sensitivity and specificity [163].

Verdonck and co-workers investigated 19 breast sections from 13 patients, through FTIR microspectroscopy (on BaF_2_ slides), achieving good discrimination between lymphocytes, connective tissue, epithelial cells, erythrocytes and vascular tissue. In addition, different stroma biochemical compositions surrounding the tumour were characterised, which enabled the authors to identify the tumour margins with >90% specificity and >80% sensitivity (except for the vascular tissue, for which a 74% sensitivity was obtained) [164].

Using breast tissue microarrays (TMA) mounted on CaF_2_, Lazaro-Pacheco and colleagues reported the FTIR analysis of 245 human breast tumours and 37 healthy sections, reaching 92% sensitivity and 86% specificity when differentiating these two types of samples [165]. Using a similar approach, Pounder et al. successfully probed 34 healthy sections, one ILC and 30 IDC breast TMA cores (on BaF_2_ slides) [166]. Different classification models were built, able to identify both tumour vs. non-tumour and stroma against epithelium. This strategy allowed the authors to quantify the epithelium content and distribution along its histological category—healthy or tumourigenic. False-coloured images were generated upon robust classification of normal vs. cancerous structures (area under the ROC (receiver operating characteristic) curve >0.9 for both transmission and reflection acquisition modes).

Additionally, using breast TMA samples coated on standard histological glass slides, Bassan et al. were able to successfully acquire microFTIR spectra in transmission mode (solely accessing the high wavenumber region) and identify epithelium, stroma, blood and necrosis based on specific spectral biomarkers, with a correct classification of 98.25%, 99.94%, 100% and 97.22%, respectively [21]. False-coloured images were obtained and malignant vs. non-malignant spectral features were identified, thus providing a faster and reliable alternative to histopathological analysis of TMAs. More recently, Tang and co-workers applied a similar approach to 120 H&E-stained breast TMA cores [167]. Cancerous vs. healthy stroma and epithelium were the tissue type categories under study for which the classifier achieved >90% accuracy for cancerous and healthy stroma, while 73% and 88% accuracy was attained for the cancerous and healthy epithelium, respectively. The final diagnosis of each core was then determined based on the number of spectra corresponding to healthy or cancerous pixels, providing accurate results in >95.8% of the cores (Figure 11).

#### 3.4.2. Spectral Histopathology of the Head and Neck

The study of 72 oral cancer FFPE samples from 57 patients was conducted by Ibrahim and co-workers on a microRaman spectrometer. After the single use of digital dewaxing, the samples were processed and categorised according to the classes benign, mild, moderate or severe dysplasia and SCC [168]. Good sensitivity values were attained for benign and SCC, but poor specificity was reached. Inflammation and smoking were also evaluated, achieving an accuracy of 94% and 76%, respectively. Likewise, the analysis of 17 FFPE tongue samples (chemically dewaxed prior the spectroscopic analysis), belonging to eight patients, diagnosed as normal, carcinoma in situ and invasive squamous cell carcinoma was reported by Devpura et al. [169]. Carcinoma in situ and normal tissue were identified with 91% success rate, and invasive squamous cell carcinoma was classified with 89% accuracy.

The use of oral squamous cell carcinoma TMAs from 14 patients was reported by Pallua and collaborators, who mounted the samples on CaF_2_ slides for FTIR imaging acquisition [170]. Biochemical images were generated by the integration of some IR bands assigned to phosphate, phospholipids and nucleic acids, as well as through k-means clustering and hierarchical cluster analysis. Although the images matched the H&E cuts, the authors did not study the discriminatory potential of their methodology to assess normal vs. cancerous spectral features.

The simultaneous use of spectral histopathology and standard histopathology reduces the subjectivity of the diagnoses, decreasing the occurrence of diagnostic errors. Furthermore, since it is possible to clearly distinguish benign from malignant lesions, vibrational spectroscopy can significantly reduce the pathologists’ workflow, allowing these professionals to focus on clinically relevant cases (the malignant ones). Ideally, spectral histopathology should be performed on stained samples in order to fully integrate this technology in the pathology laboratory routine, which Tang et al. [167] did with breast TMAs, as mentioned in Section 3.4.1, following successful work from Pilling and colleagues, who diagnosed prostate cancer in H&E slides with over 95% accuracy using infrared spectroscopy [171].

Table 1, Table 2 and Table 3 summarise the research studies included in this review for Raman, FTIR and spectral histopathology, respectively.

## 4. Discussion

Breast, and head and neck cancers are the types of neoplasias for which early diagnosis and surgical margin assessment can be largely improved. As discussed in this review paper, great research effort has been dedicated to the development of vibrational spectroscopy techniques to successfully answer these urgent clinical needs. 

Raman spectroscopy, in its several configurations, has been extensively used in cancer research. Technical improvements regarding the Raman signal intensity over tissue fluorescence, the detector’s efficiency and the development of suitable accessories (e.g., optical fibre probes) enabled the successful use of Raman for a fast, accurate and non-invasive in situ diagnosis, applicable both during screening and intraoperatively. The combined effort of research facilities, clinical centres and the industry will hopefully allow the translation of Raman spectroscopy to clinics in the near future, based on leading-edge technology successfully developed in the past few years.

FTIR spectroscopy, in turn, faces some other challenges regarding fresh tissue analysis due to water content. The use of ATR configurations may be the solution to overcome water spectra “contamination”. Additionally, very accurate data were obtained envisaging an early diagnosis from dried blood serum, saliva and cell pellet samples. The use of these types of samples is especially advantageous, since their collection is easy, non-invasive and fast, rendering FTIR-ATR an excellent candidate as a routine cancer screening methodology, particularly for some types of SCC (e.g., HNC). 

It is presently widely recognised that spectral histopathology is able to distinguish benign from cancerous tissue features. In an early stage, spectral histopathology could be used as a screening method to assist anatomopathologists by allowing them to focus only on the suspicious samples without the need to observe those classified spectroscopically as benign. Secondly, spectral histopathology may complement the anatomopathologists’ interpretation of the specimens by accurately distinguishing the pre-malignant lesions, thus overcoming the subjectivity associated with histopathology practice. In order to attain this goal, extensive work is still needed, mainly regarding the identification of the intermediate stages between benign and malignant features. Hopefully, when this lacuna is finally fulfilled, spectral histopathology will become an invaluable clinical tool for early cancer diagnosis and intraoperative margin assessment, greatly contributing to an improved prognosis of oncology patients.

It is important to note that vibrational data need extensive data pre-processing and processing work prior to analysis and subsequent final diagnosis, which may be more time-consuming than the acquisition of the data themselves. Spectral data need to be background subtracted, baseline corrected, signal-to-noise checked, smoothed, artifact and interferent cleaned (e.g., glass, paraffin, light-scattering corrections), and normalised prior data selection, for subsequent outlier detection and modelling. Modelling can be developed with PCA, discriminant analysis (e.g., linear, quadratic), PLS, SVM, random forest, hierarchical cluster analysis, k-nearest neighbour or a combination of two or more of these statistical and deep-learning algorithms, which are some examples of the most used approaches for the data analysis with diagnostic purposes. Afterwards, validation of the developed model is needed. Additionally, there is a lack of a recognised universal protocol for vibrational data acquisition, pre-processing, processing and analysis, so these technologies can be finally applied in a clinical setting. However, some review works have been published aiming at the standardisation of the protocols for vibrational diagnosis [172,173,174,175].

## 5. Conclusions

In this literature review, Raman and FTIR research works aiming at breast and head and neck cancer early diagnosis, surgical margin assessment and spectral histopathological diagnosis were covered. Highly innovative, accurate, sensitive and specific methodologies were included, confirming the promising potential of vibrational spectroscopy in the clinical workflow.

## Figures and Tables

**Figure 1 cancers-13-05336-f001:**
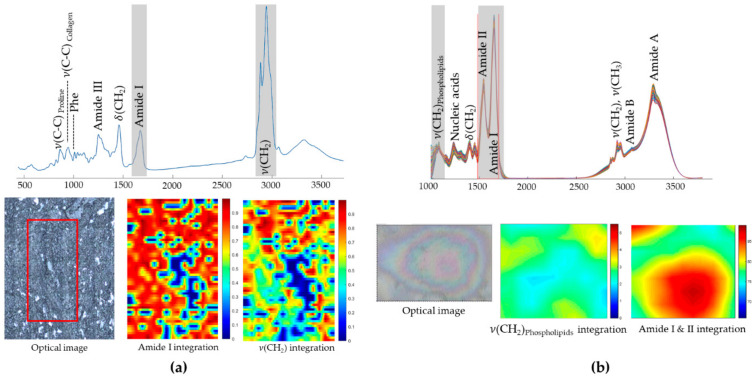
Typical spectra and biochemical images obtained by Raman and infrared microspectroscopy of biological samples: (**a**) Raman microspectroscopy of a human breast tissue section (Material not intended for publication: Mamede, A. P., Santos, I. P., Batista de Carvalho, L. A. E., QFM-UC, Coimbra, Portugal. Biochemical image of amide I and CH_2_ stretching distribution in a human breast section, 2021); (**b**) infrared microspectroscopy (with synchrotron radiation) of a human osteosarcoma cancer cell (MG-63) [85].

**Figure 2 cancers-13-05336-f002:**
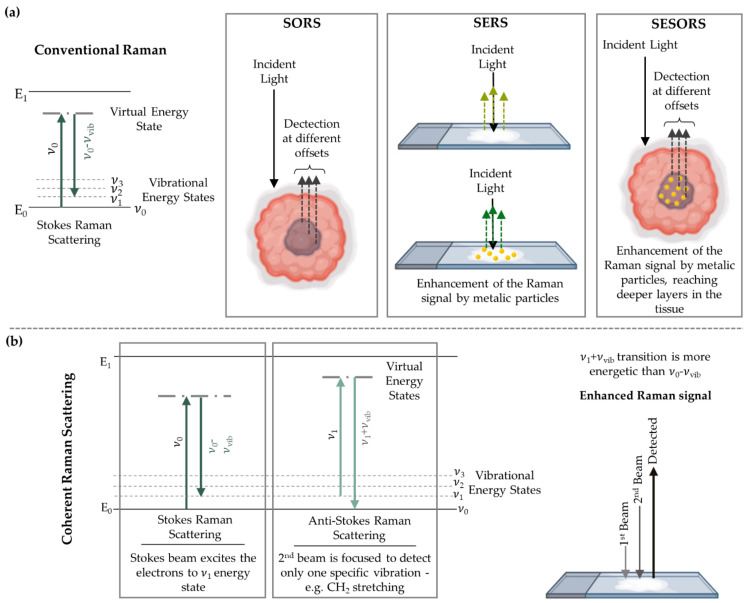
Schematic representation of several Raman configurations: (**a**) conventional Raman, SORS, SERS and SESORS; (**b**) coherent Raman scattering (CRS).

**Figure 3 cancers-13-05336-f003:**
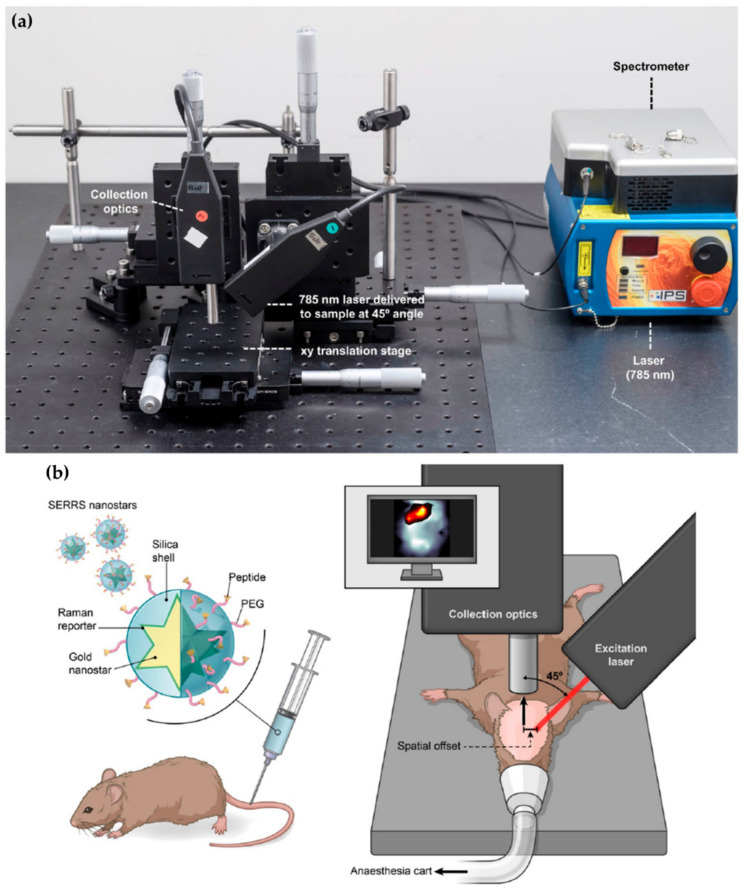
(**a**) SORS setup used by Nicolson et al. in the study of glioblastoma multiforme tumours; (**b**) conceptual scheme outlining the integrin-based detection of GBM through the use of cRGDyK-conjugated SERRS nanostars and in vivo SESORRS imaging of GBM performed in a custom-built SORS system depicted in (**a**) [101].

**Figure 4 cancers-13-05336-f004:**
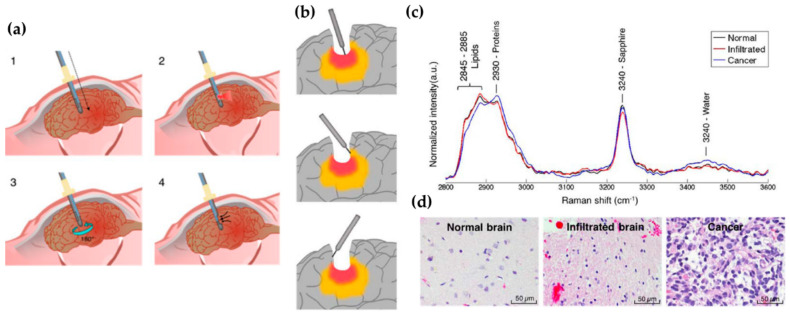
Schematic representation of (**a**) the acquisition steps of the Raman core needle biopsy instrument developed by Desroches et al.: (1) needle insertion, (2) Raman data acquisition, (3) 180° rotation of the needle, and (4) sample collection. (**b**) In vivo Raman measurements performed in the surgical cavity during glioma resection in dense cancer (red), infiltrated brain (yellow) and surrounding normal brain. (**c**) Normalised average Raman spectra of dense cancer, infiltrated and normal brain. (**d**) Representative H&E micrographs for each tissue type [103].

**Figure 5 cancers-13-05336-f005:**
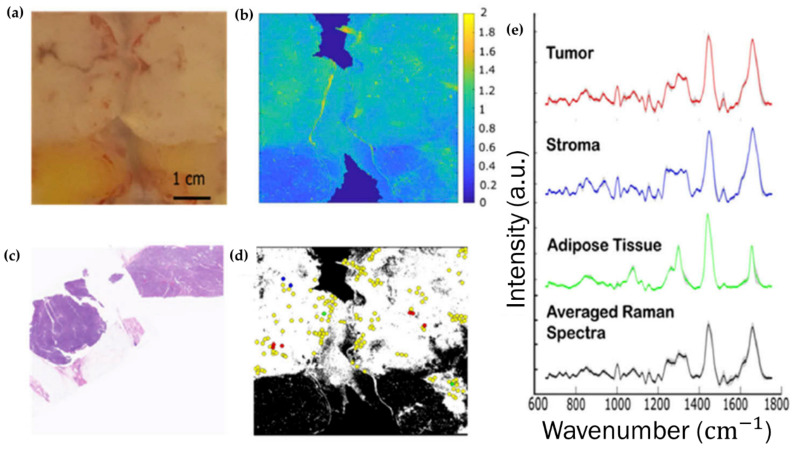
Combined TIR-AF and Raman spectroscopy analysis of a human specimen containing a phyllodes tumour. (**a**) optical image; (**b**) ratiometric TIR-AF image; (**c**) H&E-stained tissue section; (**d**) threshold image (0.8 < T < 2), showing the sampling locations for the Raman acquisition, dotted in yellow; (**e**) average Raman spectra of stroma (blue), tumour (red) and adipose tissue (green) and corresponding standard deviation (shaded in grey) [109].

**Figure 6 cancers-13-05336-f006:**
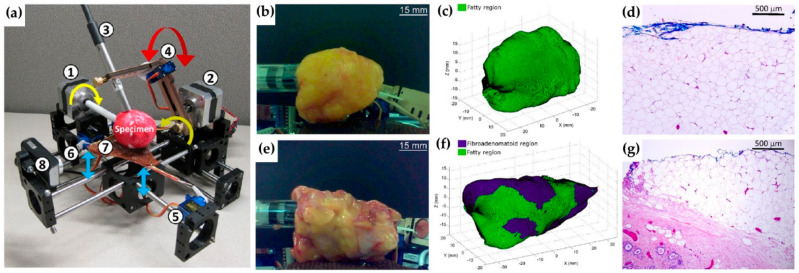
(**a**) Photograph of the automated 3D margin scanner prototype (Marginbot), developed by Thomas et al.: ① motor A rotates the specimen in the horizontal axis, while ② motor B moves the ③ optical probe along the specimen’s surface; ④ servomotor A enables the contact mode and non-contact mode of the optical probe ①with the specimen placed on the ⑦ specimen holder; ⑤, ⑥ servomotor B moves the specimen holder ⑦ upwards, pushing the specimen towards the probe during the contact mode and downwards in the non-contact mode. The (⑧) compact camera enables image reconstruction of the specimen. (**b**–**g**) Automated 3D margin assessment of human breast specimens. (**b**) Photograph of the breast specimen with fatty margins; (**c**) margins of the specimen rendered by the scanner; (**d**) 10× magnification of an H&E human breast section; (**e**–**g**) corresponding figures for a breast specimen with fibroadenomatoid margins (color code for margin classification in (**c**) and (**f**): green—>50% fat composition, blue—>50% fibroepithelial/fibro-glandular composition) [111].

**Figure 7 cancers-13-05336-f007:**
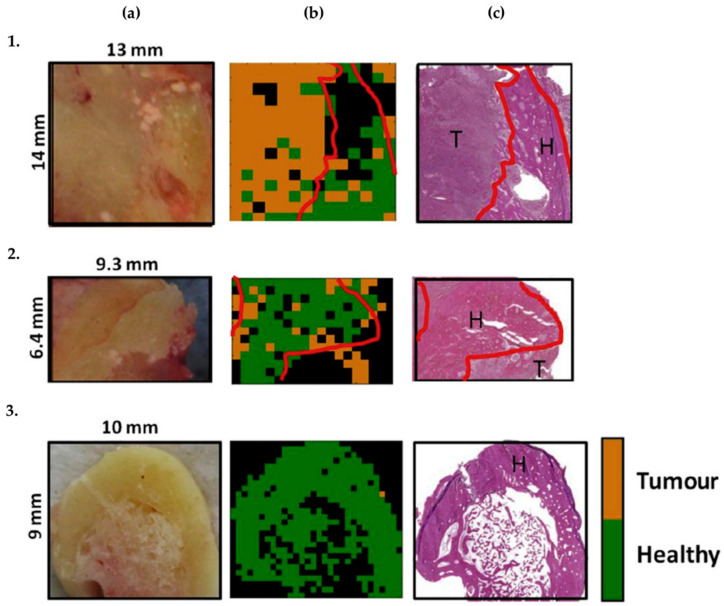
Examples of the results obtained when testing the tissue classification model developed by Barroso et al. (**a**) Photographs of fresh bone cross sections from the centre of a resection specimen with tumour invasion (based on radiologic examination) from which Raman spectra were obtained. (**b**) Raman classification maps (orange: tumour, green: healthy bone) tumour border indicated in red. Black pixels correspond to the absence of tissue or to spectra with low Raman signal quality. (**c**) H&E-stained sections obtained from the measured bone surface, with tumour border (red), tumour (T) and healthy surrounding bone (H) indicated by the pathologist, obtained from 3 bone slices from 3 patients (1 to 3) [139].

**Figure 8 cancers-13-05336-f008:**
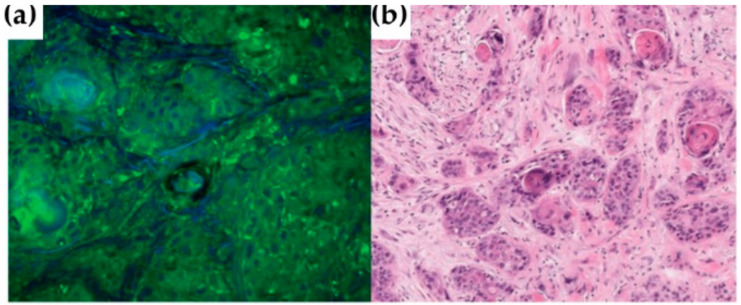
(**a**) CRS microscopy image generated through the contrast between CH2 and CH3 stretching vibrations at 2845 cm^−1^ and 2940 cm^−1^, respectively, demonstrating hypercellu-larity and atypical nuclei, obtained by Hoesli et al. [121] and (**b**) Corresponding H&E stained specimen [142].

**Figure 9 cancers-13-05336-f009:**
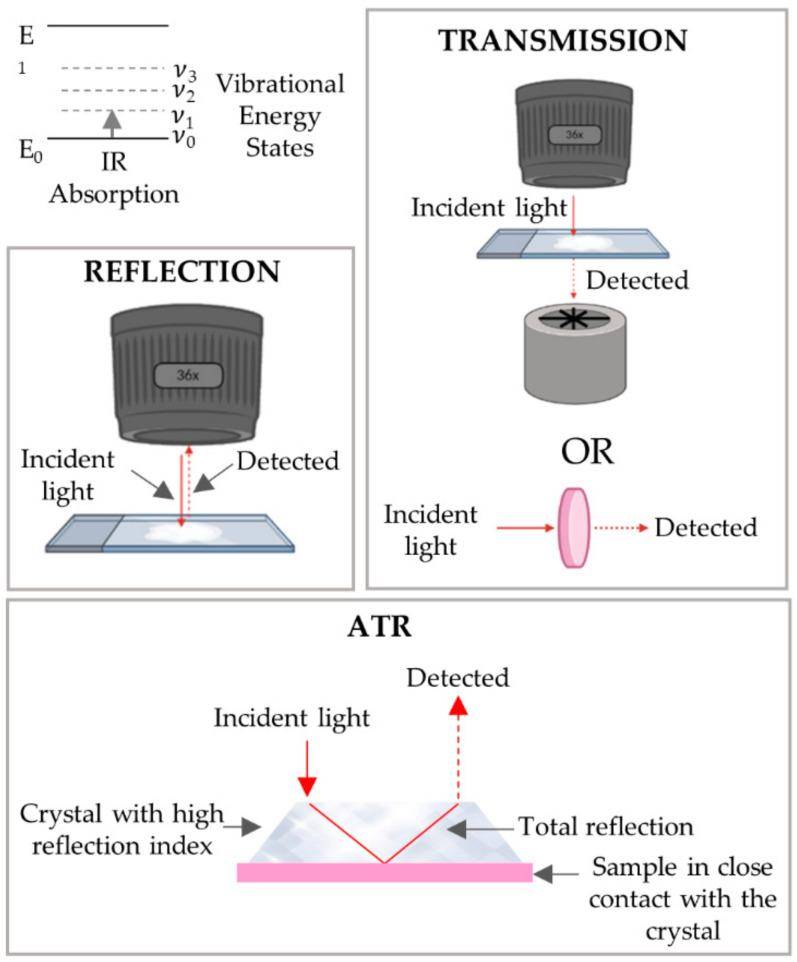
Schematic representation of the energetic transition occurred during infrared absorption and FTIR modes of acquisition: transmission, reflection and microATR.

**Figure 10 cancers-13-05336-f010:**
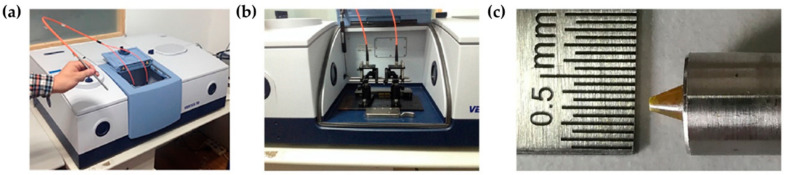
(**a**) Photo of the ATR-HOF probe developed by Zhao et al.; (**b**) fixing-coupling equipment used to attach the probe to the spectrometer and (**c**) end of the ATR-HOF crystal probe [148].

**Figure 11 cancers-13-05336-f011:**
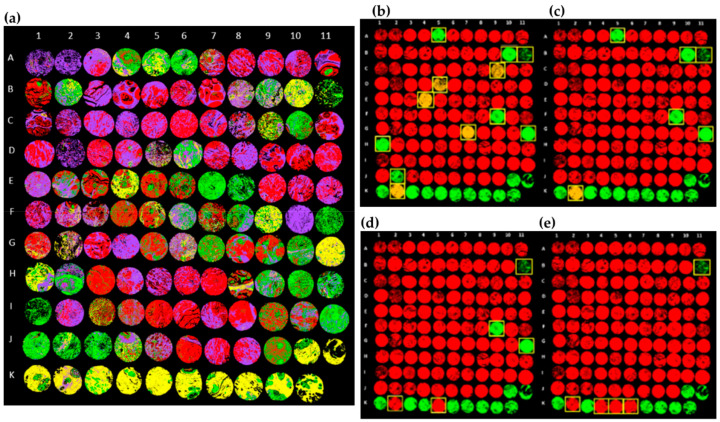
(**a**) False-coloured images of 120 grade II breast cancer cores: red—cancerous epithelium, purple—cancerous stroma, green—normal associated epithelium and yellow—normal associated stroma; (**b**–**e**) false-coloured images of 120 grade II breast cancer cores using a traffic light system with a threshold of: (**b**) 0.5, (**c**) 0.3, (**d**) 0.2, (**e**) 0.1, where red represents cancerous cores, green represents normal cores and amber represents cores that could not be classified as either cancerous or normal, and therefore should be looked at by a pathologist. The yellow boxes indicate cores which were wrongly classified [167]. Alterations made to original Figure 5: (**c**,**d**) placed parallel to (**a**,**b**).

**Table 1 cancers-13-05336-t001:** Summary of the research studies included in this review article, performed with Raman spectroscopy, according to the type of sample, sample preparation, methodology and specifications used.

Early Diagnosis								
Reference	Sample	Sample Preparation	Methodology	Depth	Specificity (%)	Sensitivity (%)	Accuracy (%)	Observations
Cervo et al. [93]	Human blood serum (BC patients)	-	SERS		85	92	90	Localised and locally advanced breast cancer distinguished with 84% accuracy
Nargis et al. [94]	Human blood serum (BC patients)	-	SERS	-	100	99	-	-
Nargis et al. [95]	Human blood serum (BC patients)	-	SERS	-	98.4	90	-	
			microRamn		97.7	88.2		
Moisoiu et al. [96]	Human blood serum (cancer patients)	-	SERS	-	93.6	93.7	-	BC was differentiated from lung, colorectal, oral and ovarian cancer with 76% accuracy
Moisoiu et al. [97]	Human urine (BC patients)	-	SERS	-	95	81	88	-
Lin et al. [98]	Human urine (cancer patients)	Filtration through affinity chromatography	SERS	-	87.5	76.5	-	BC was distinguished from gastric cancer with 82% sensitivity and 90.7% specificity
Nicolson et al. [99]	Breast cells	Incubation with nanotags	SESORRS	10 mm	-	-	-	3D multicellular cell spheroids
Nicolson et al. [100]	Breast cells	Incubation with nanotags	SESORRS	15 mm	-	-	-	3D multicellular cell spheroids
Nicolson et al. [101]	Mice brain	Injection of Functionalised nanoparticles	SESORS	Through the mice skull	-	-	-	Animal models, in vivo
Desroches et al. [103]	Human brain	Intraoperatively	Adaptation of core-needle instrument	-	90	80	84	Clinical trial, in vivo
Saha et al. [104]	Human breast	Fresh	Portable spectroscopy system	-	97	77	-	-
Barman et al. [106]	Human breast	Fresh	Portable spectroscopy system	-	-	-	82	
Connolly et al. [117]	Saliva from HNC patients	-	SERS	-	57	89	-	-
	Oral cells from HNC patients				52	68	-	
Falamas et al. [118]	Saliva from HNC patients	Liophilised	microRaman	-	-			Seven bands with discriminatorypotential were identified with 83% accuracy.
Sahu et al. [119]	Blood serum (HNC patients)	-	microRaman	-	-			78% efficiency
Xue et al. [120]	Blood serum (HNC patients)	-	SERS	-	-			T1 and T3 OSCC distinguished with 80% accuracy, T2 and T4 with 71.1% and 77.8% accuracy, respectively. N0, N1 and N2 distinguished with 75.5%, 93.8% and 92%, respectively.
Brindha et al. [121]	Human urine (HNC patients)	-	microRaman	-	87.1	98.6	93.7	-
Brindha et al. [122]	Human urine (HNC patients)	-	microRaman	-	98.7	91.9	-	Only the high wavenumber region was used. Achieving higher classification accuracies.
Jaychandra et al. [123]	Human urine (HNC patients)	-	microRaman	-	-	-	90.5	Malignant, pre-malignant and healthy samples were tested
	Human saliva (HNC patients)				-	-	93.1	
	Blood serum (HNC patients)			-	-	-	78	
	Biopsy-collected tissue			-	-	-	97.4	
Singh et al. [124]	HNC patients	In vivo	Fire-optic probe coupled to a spectrometer	-				Healthy controls, healthy tobacco users and pre-malignant samples were correctly predicted with 94%, 86% and 55% efficiency, respectively
Krishna et al. [125]	HNC patients	In vivo	Portable Raman device	-	94	94	-	Pre-malignant lesions were correctly classified in 88% of the measured sites and malignant lesions in 84% of the sites.
Guze et al. [126]	Oral abnormalities	In vivo– mimicking a clinical setting	Raman probe	-	77	100	-	-
Bergholt et al. [128]	Transnasal tissues	In vivo	Raman probe–possible integration with endoscopes	-	-			The spectra from different areas of the head and neck obtained and characterised.
Vohra et al. [129]	RNA from HNSCC	Fresh	SERS	-	89	100	-	-
	RNA from Lymph nodes							
Sahu et al. [130,131,132,133]	Exfoliated cells (HNC patients)	Cell pellets, placed on CaF_2_	Fibre optic Raman microprobe	-	-	70	-	A spectra-wise analysis of the data provided a sensitivity of 77%
Hole et al. [134]	Exfoliated cells (HNC patients)	Cell pellets, placed on CaF_2_	Fibre optic Raman microprobe	-	-	-	-	Classification efficiency of 86%
Ghosh et al. [135]	Exfoliated cells (HNC patients)	Suspension of cells	microRaman	-	-	-	80	A spectra-wise analysis of the data provided a sensitivity of 82%
Behl et al. [136]	Exfoliated cells (HNC patients)	Monolayer of cells placed on glass slides	microRaman	-	-	94	-	Analysing the cells’ nuclei
				-	-	86	-	Analysing the cells’ cytoplasm
**Surgical Margins Assessment**								
**Reference**	**Sample**	**Sample preparation**	**Methodology**	**Acquisition time**	**Specificity (%)**	**Sensitivity (%)**	**Accuracy (%)**	**Observations**
Koya et al. [106]	Human breast	Frozen sections	microRaman	-	90.8	88.8	90	-
Kong et al. [107]	Human breast	Frozen sections	microRaman	-	96.2	95.6	-	
			MSH	17 min				
Shipp et al. [108]	Human breast	Fresh	MSH	12–24 min	82	95	-	-
Lizio et al. [109]	Human breast	Fresh	MSH (using TIR-AF)	45 min (possible reduction to 20 min)	-			H&E in agreement with Raman heatmaps.
Liao et al. [110]	Human breast	Fresh	Altered Raman spectrometer	20–25 min	-			H&E in agreement with Raman heatmaps.
Thomas et al. [111]	Human breast	Fresh	Raman 3D scanner	7–15 min	85	93	-	-
Keller et al. [112]	Human breast	Frozen	SORS probe	-	100	95	-	
Wang et al. [113]	Human breast	Fresh	SERS	10–15 min	92.1	89.3	-	
Horsnell et al. [114]	Human lymph nodes (breast surgery)	Frozen	microRaman	-	97	81	-	
Petterson et al. [115]	Human lymph node	Fresh	Multifibre Raman probe in a hypodermic needle	Some seconds	-			Spectra with good signal to noise obtained.
Zúñiga et al. [116]	Human breast	Frozen	Portable and commercially available Raman devices	-	-	-	> 90	-
Barroso et al. [137]	SCC of the tongue	Fresh	Built in Raman equipment	Within 30 min after excision	92	99	-	
Barroso et al. [138]	Oral SCC	Fresh	Built in Raman equipment	Within 30 min after excision	-			Water concentration in the tumour was found to be 76% ± 8% and, 56% ± 24% in the surrounding healthy tissue
Barroso et al. [139]	Oral SCC—bone	Fresh	Built in Raman equipment	<30 min	87	95	95	-
Cals et al. [140]	Oral SCC	Frozen	Inverted microRaman	Within 60 min after excision	-	-	97	
Cals et al. [141]	SCC of the tongue	Frozen	Inverted microRaman	Too long for clinical application	78	100	91	
					66	100	86	
Hoesli et al. [142]	Human head and neck	Frozen	CRS	-	-			H&E in agreement with Raman heatmaps

**Table 2 cancers-13-05336-t002:** Summary of the research studies included in this review article, performed with FTIR spectroscopy, according to the type of sample, sample preparation, methodology and specifications used.

Early Diagnosis							
Reference	Sample	Sample Preparation	Methodology	Specificity (%)	Sensitivity (%)	Accuracy (%)	Observations
Ferreira et al. [143]	Saliva (BC patients)	Lyophilised	ATR	70	90	-	-
Zelig et al. [144]	Blood serum and peripheral mononuclear cells (BC patients)	Air dried on ZnSe slides	microFTIR	78	87		
Elmi et al. [145]	Blood serum (BC patients)	Dry at room temperature	ATR	85	92	90	
Sitnikova et al. [146]	Blood serum (BC patients)	Dry at room temperature	ATR	87.1	92.3	-	
Mastanduno et al. [147]	BC patients	In vivo	MRI-NIRS	-			Successful identification of malignant apart from benign lesions.
Zlotogorski-Hurvitz et al. [152]	Exossomes from the saliva of HNC patients	Dry at room temperature	ATR	89	100	95	-
Falamas et al. [118]	Saliva from HNC patients	Dry on KBr tablets	FTIR, transmission mode	-			Three main signals found to be biomarkers with 82% accuracy.
Rai et al. [153]	Bold serum from OSF patients	Dried under vacuum	TR-FTIR	-			Predictive capability and explained variance higher than 0.9.
Townsend et al. [154]	Exfoliated cells (HNC patients)	Cell pellets, placed on low-*e* slides	microFTIR, in reflection mode	>90			
Ghosh et al. [135]	Exfoliated cells (HNC patients)	Suspension of cells	ATR			>80	The integrated approach of both Raman and FTIR provided an overall accuracy of 97.7%
**Surgical Margins Assessment**							
**Reference**	**Sample**	**Sample** **preparation**	**Methodology**	**Specificity (%)**	**Sensitivity (%)**	**Accuracy (%)**	**Observations**
Lu et al. [149]	Human breast	Fresh	ATR-HOF	-	-	>90	-
Tian et al. [150]	Sentinel lymph nodes from BC patients	Fresh	ATR-HOF	90.1	94.7	-	
Tian et al. [151]	Human breast	Fresh	ATR-HOF	98	90	-	

**Table 3 cancers-13-05336-t003:** Summary of the research studies included in this review article, performed with Raman and FTIR spectroscopy regarding spectral histopathology, according to the type of sample, sample preparation, methodology and specifications used.

Spectral Histopathology							
Reference	Sample	Sample Preparation	Methodology	Specificity (%)	Sensitivity (%)	Accuracy (%)	Observations
Vanna et al. [162]	Human breast	Chemical dewax	microRaman	80.6	93.5	-	-
Lyng et al. [163]	Human breast	Chemical dewax	microRaman	>90	>90	-	
Verdonck et al. [164]	Human breast	Chemical dewax, on BaF_2_ slides	microFTIR	>90	>80	-	
Lazaro-Pacheco et al. [165]	Human breast	Chemical dewax, on CaF_2_ slides	microFTIR	86	92	-	
Pounder et al. [166]	Human breast	Chemical dewax, on BaF_2_ slides	microFTIR	-			Area under the ROC curve >0.9 for both transmission and reflection acquisition modes
Tang et al. [167]	Human breast	H&E stained, on standard histological glass	microFTIR	-	-	95.8	Cancerous vs. healthy stroma classified with >90% accuracy; healthy epithelium and cancerous epithelium classified with 73% and 88% accuracy, respectively.
Bassan et al. [21]	Human breast	Chemical dewax, on standard histological slides	microFTIR	-			Epithelium, stroma, blood and necrosis correctly classified in 98.25%, 99.94%, 100% and 97.22%, respectively, of the cases
Ibrahim et al. [168]	Human head and neck	FFPE samples, digitally dewaxed	microRaman	-			Inflammation and smoking factors successfully classified with 94% and 76% accuracy, respectively.
Devpura et al. [169]	Human head and neck	Chemical dewax	microRaman	-	-	89	-
Pallua et al. [170]	Oral SCC	Chemical dewax, on CaF_2_ slides	microFTIR	-			H&E in agreement with FTIR heatmaps

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
