# Peer review of "A New Look into Cancer—A Review on the Contribution of Vibrational Spectroscopy on Early Diagnosis and Surgery Guidance"

_cancers, 2021, doi:10.3390/cancers13215336_

Round 1
Reviewer 1 Report
The manuscript is a review about vibrational spectroscopy contributions on early diagnosis and surgery guidance.
The work is fundamentally sound: the literature review was thorough, possibilities and limitation of the considered methodologies have been clearly presented and discussed and key concept have been clearly highlighted.
I thus recommend the publication of the manuscript in its actual form.
Author Response
The authors kindly appreciate the reviewer´s positive evaluation and comments. Nevertheless, two more reviewers have requested major revisions to the manuscript. Hence, the paper was reorganized according to the PRISMA checklist, and further research studies were added regarding the early diagnosis of breast and head and neck cancers. The authors hope that referee 1 agrees with this new version.
Looking forward to a positive recommendation for publication, with my best regards,
Adriana P. Mamede
Reviewer 2 Report
The work “A New Look into Cancer – A Review on the Contribution of Vibrational Spectroscopy on Early Diagnosis and Surgery Guidance”, by A.P. Mamede et al., reviews many articles about the application of Raman and FTIR techniques as diagnostic tools to differentiate healthy and cancer biological samples, both in vitro and in vivo conditions. Such a review is made of i) two introductive sections (1 and 2), in which the main techniques currently used to detect the carcinogenicity of breast and oral tissues for diagnosis and surgical removal purposes are cited and explained, as well as their biochemical and technical drawbacks and limitations, ii) a short description of the basic theory and the operating modalities of the two vibrational techniques, iii) a very synthetic review of some literature results about the use of vibrational techniques for early detection of breast and oral cancers, mainly based on the analysis of biological fluids, iv) a proper review about the use of the vibrational spectroscopy in clinical operating room for surgical margins assessments of breast and oral cancer tissues, v) an additional section about the use of Raman and FTIR spectroscopy to obtain spectral maps of the biochemical content of human tissues, which can be complementary to histopathological images and vi) a conclusions section about the reviewed results.
The reviewed topic is interesting because vibrational spectroscopies can provide, in a label-free way, biochemical information and images for clinical diagnosis. In fact, several literature papers proposed such techniques as a complementary tool to support anatomopathologists to make diagnostic reports.
The review about surgical margins assessment is quite wide and the literature articles used for writing the manuscript are appropriate to the length of the review. On the contrary, the description of the early diagnosis possibilities is too concise and lacking an adequate number of literature references. Therefore, the work might be suitable to publication in Cancers only after several issues have been properly addressed and improved. Such points are described in the following.
Major revisions
- Suggestion: sections 1 and 2 can be merged into a single introductory section, at the end of which vibrational spectroscopic techniques could be proposed as complementary and support methods for clinical diagnostics.
- Page 7: references for SERS, SESORS, SESORRS and CRS techniques should be provided, particularly as for the theoretical aspects is concerned.
- The review about the diagnostic capabilities of vibrational spectroscopies is insufficient. The authors briefly report about diagnostics carried out using biological fluids, whereas they don’t review about diagnostics carried out for human tissues and cells. Therefore, if the authors consider it appropriate to keep such a diagnostics part, they should broadly expand it, both in the discussion and in the bibliographic references relating to histological (tissue, biopsy) and cytological samples.
- In fact, many reviews about diagnostics possibilities of vibrational techniques by using biological fluids, cells and tissues (including the breast and oral cavity) have been previously published. Instead, I think the part relating to surgical margins assessment is newer and more interesting than that about diagnostics. That is, you could also leave this part exclusively, if the authors think it appropriate (also changing the title of the review).
- In most cases, vibrational techniques are unable to provide useful information without adequate pre-treatment of the measured spectra and the aid of multivariate analysis techniques. Obviously, this increases the time within which usable information can be obtained (compared to the time required to carry out the measurements). This aspect should be included and emphasized in the review, also specifying what are the analysis techniques most reported in the published papers about vibrational spectroscopy of biological samples. In fact, the use of few multivariate analysis techniques was only mentioned in the review at pages 15 and 18.
- The section 3.3 should also include few Raman and/or FTIR images of breast and oral tissues from the cited literature papers. In addition, some references about the chemical and digital dewaxing techniques should be added.
- A major limitation in the use of vibrational techniques for diagnostic purposes is the absence of a universally recognized protocol about the methods of measurement, the pre-treatment of the spectra and the analysis of data. Some review works have been published in order to overcome this limitation (Baker, M., Trevisan, J., Bassan, P. et al.Using Fourier transform IR spectroscopy to analyze biological materials. Nat Protoc 9, 1771–1791 (2014); Morais CLM, Lima KMG, Singh M, Martin FL. Tutorial: multivariate classification for vibrational spectroscopy in biological samples. Nat Protoc. 2020 Jul;15(7):2143-2162; Butler, H., Ashton, L., Bird, B. et al. Using Raman spectroscopy to characterize biological materials. Nat Protoc 11, 664–687 (2016)]. These aspects should be included and discussed in the review, at least in the conclusions sections.
Minor revisions
- Typo at line 62: please correct “sise” into “size”.
- Line 275: please correct “light wave and the atoms” into “light wave and electric charges inside atoms”.
- Lines 280-281: please correct “the energies at which the atoms’ displacements occur” into “the energies of vibrational states.
- Line 285: please correct “slight displacements when ther is a different” into “slight shifts according to different”.
- Line 292: Figure 1a is correct for Raman.
- Line 301: “rough nanostructured metal surface”.
- Figure 4c: have the average spectra been normalized? Please specify in the caption.
- Line 610: please insert the exact section and figure numbers for the work of Hoesli et al.
- Line 729-730: such a sentence can be removed, because the content has been specified in the previous sub-section 3.2.3.
Author Response
The authors thank reviewer 2 for his/her careful reading and evaluation of the submitted manuscript. All suggestions were taken into account, since the authors believe they will render this review paper more complete and valuable.

Reviewer 3 Report
Dear authors,
the article concerns an interesting topic. Yet, it is not organized as a scientific paper but more like a book chapter/essay. Independently from the fact that the things you have written may be right or interesting, reviews should be managed as scientific papers, with proper methods, specifing what type of review you have performed (systematic, scoping, with/without metanalysis, etc...) and how you performed it (keywords, search engines, bias risk assessment, etc...). I suggest you follow PRISMA check-list.
In my opinion authors should re-arrange it as a proper review and resubmit.
Kind regards
Author Response
As suggested, the manuscript was reorganized according to the PRISMA checklist. A section dedicated to the methodology was added, apart from Results, Discussion and Conclusions.
Looking forward to a positive recommendation for publication, with my best regards,
Adriana P. Mamede
Round 2
Reviewer 2 Report
The manuscript has improved from the previous version, after the revisions made. In particular, I appreciated the inclusion of the "Methods" section.
My only doubt about this revised version concerns the keywords reported on lines 108-111 and 113-117. In my opinion they are superfluous and do not add important information to the work. These keywords should be removed.
Author Response
The authors thank Reviewer 2 for the positive evaluation on the revisions made to the manuscript. Regarding the keywords, they were kept since referee 3 considered them to be relevant, although changed according to search query (use of OR and AND).
With my best regards,
Maria Paula Matos Marques
Reviewer 3 Report
Dear authors,
the article is improved, but methods are still inadequate in my opinion, as you have not followed the PRISMA statement if not very superficially.
a) First of all, if you have followed the PRISMA statement, you have to specify it in the methods. Furthermore, PRISMA checklist should be added to additional files in order to understand where each point has been discussed.
b) I advise against the use of Google Scholar, as it does not filter for predatory journal. Scopus and PUBMED are sufficient. Rearrange results accordingly.
c) Refine your search methods; [line 108 "The keywords used for the search were “breast cancer”...]: this is not a proper search string. Search query should be written properly, e.g.: "Research query was: (TITLE-ABS-KEY (“breast cancer”) AND/OR TITLE-ABS-KEY (“screening”)) etc....". "Elsevier" is not the name of the search engine, but a publishing house owner of the "SCOPUS" search engine; you have to be more precise about what search engine you used.
d) "[LINE 113] The filtering process included at least three of the following keywords": this passage should be clarified, as suggested by the PRISMA statement.
e) Most of the PRISMA statement checklist is missing: no flow diagrams, no risk of bias assessment, and so on. Add the PRISMA statement in the additional file, after having followed it thoroughly.
Conclusively, methods are still inadequate and the paper should be re-evaluated afted major revision.
Kind regards
Author Response
The authors thank Reviewer 3 for such important comments. The re-organization of the manuscript according to: Introduction, Methods, Discussion and Conclusions is indeed very useful, especially the inclusion of the Methods sections, in our opinion significantly improved the overall quality of the manuscript. Hence, this request was answered in a first revision.
The authors answered to the referee´s suggestions, points (b), (c) and (d) – changes marked in red along the text.
However, we would like to emphasize that this not is intended as a systematic review, but rather as a literature review – covering the latest progress regarding the application of optical vibrational spectroscopy techniques to the early diagnosis and surgical margins assessment of breast and head and neck cancers. Therefore, the PRISMA guidelines are not compulsory.
With my best regards,
Maria Paula Matos Marques